



# Significant impact of urban-tree biogenic emissions on air quality estimated by a bottom-up inventory and chemistry-transport modeling

Alice Maison[1,2], Lya Lugon[1], Soo-Jin Park[1], Alexia Baudic[3], Christopher Cantrell[4], Florian Couvidat[5], Barbara D'Anna[6], Claudia Di Biagio[7], Aline Gratien[7], Valérie Gros[8], Carmen Kalalian[8,a], Julien Kammer[6], Vincent Michoud[7], Jean-Eudes Petit[8], Marwa Shahin[6], Leila Simon[8,b], Myrto Valari[9], Jérémy Vigneron[3], Andrée Tuzet[2], and Karine Sartelet[1]

[1]CEREA, École des Ponts, EDF R&D, IPSL, Marne-la-Vallée, France
[2]Université Paris-Saclay, INRAE, AgroParisTech, UMR EcoSys, 91120 Palaiseau, France
[3]Airparif, Association Agréée pour la Surveillance de la Qualité de l'Air en région Île-de-France, 7 rue Crillon, 75004 Paris, France
[4]Univ Paris Est Creteil and Université Paris Cité, CNRS, LISA, F-94010 Créteil, France
[5]Institut National de l'Environnement Industriel et des Risques, Verneuil-en-Halatte, France
[6]Aix Marseille Univ, CNRS, LCE, Marseille, France
[7]Université Paris Cité and Univ Paris Est Creteil, CNRS, LISA, F-75013 Paris, France
[8]Laboratoire des Sciences du Climat et l'Environnement, CEA/Orme des Merisiers, 91191 Gif-sur-Yvette, France
[9]LMD/IPSL, École Polytechnique, Université Paris Saclay, ENS, PSL Research University; Sorbonne Universités, UPMC Univ Paris 06, CNRS, Palaiseau, France
[a]now at Université Paris-Saclay, INRAE, AgroParisTech, UMR EcoSys, 91120 Palaiseau, France
[b]now at Atmospheric Composition Research, Finnish Meteorological Institute, 00101 Helsinki, Finland

**Correspondence:** Alice Maison (alice.maison@enpc.fr), Karine Sartelet (karine.sartelet@enpc.fr)

**Abstract.** Biogenic Volatile Organic Compounds (BVOCs) are emitted by vegetation and react with other compounds to form ozone and secondary organic matter (OM). In regional air-quality models, biogenic emissions are often calculated using a Plant Functional Type approach, which depends on the land-use category. However, over cities, the land-use is urban, so trees and their emissions are not represented. Here, we develop a bottom-up inventory of urban-tree biogenic emissions, in which the

location of trees and their characteristics are derived from the tree database of the Paris city combined with allometric equations. Biogenic emissions are then computed for each tree based on their leaf dry biomass, tree-species dependent emission factors and activity factors representing the effects of light and temperature. Emissions are integrated in WRF-CHIMERE air-quality simulations performed over June-July 2022. Over Paris city, the urban tree emissions have a significant impact on OM, inducing an average increase of OM of about 5%, reaching 14% locally during the heatwaves. Ozone concentrations increase by 1.0% on

average, by 2.4% during heatwaves with local increase of up to 6%. The concentration increase remains spatially localized over Paris, extending to the Paris suburbs in the case of ozone during heatwaves. The inclusion of urban-tree emissions improves the estimation of OM concentrations compared to in situ measurements, but they are still underestimated as trees are still missing from the inventory. OM concentrations are sensitive to terpene emissions, highlighting the importance of favoring urban tree species with low terpene emissions.





## 1   Introduction

With an increasing number of people living in cities, urban areas are experiencing continuous expansion (Angel et al., 2011; United Nations, 2018). Artificial surfaces with darker, impermeable materials and high buildings, as well as release of anthropogenic heat strongly modify the energy budget of the urban area (Taha et al., 1988; Taha, 1997; Pigeon et al., 2007a; Kuttler, 2008; Oke et al., 2017; Masson et al., 2020). An increase in temperature in the city compared to the surrounding countryside

is often observed and is called Urban Heat Island (UHI) effect (Oke, 1982; Kim, 1992). Due to the high local emission sources (traffic) and the modification of air flows by tall buildings which limits the pollutant dispersion, concentrations of several pollutants, such as $NO_2$ and particles, are higher in cities than surroundings (Lyons et al., 1990; Fenger, 1999; Thunis, 2018; Li et al., 2019; Yang et al., 2020).

To mitigate the negative effects of urbanization, urban vegetation and trees in particular are now widely promoted (Livesley

et al., 2016; Chang et al., 2017; Roeland et al., 2019). Trees can reduce surrounding temperatures by creating shade and by evaporating water through transpiration (Jamei et al., 2016; Taleghani, 2018; Lai et al., 2019; Hami et al., 2019; Nasrollahi et al., 2020). Trees can also remove gaseous and particulate pollutants from the atmosphere by dry deposition, although this effect is quantitatively questionable due to the large variability and uncertainties (Nowak et al., 2006; Escobedo and Nowak, 2009; Setälä et al., 2013; Selmi et al., 2016; Nemitz et al., 2020; Lindén et al., 2023).

Trees are known to naturally emit Biogenic Volatile Organic Compounds (BVOCs). The term BVOC includes gaseous non-methane hydrocarbons and includes many families of molecules: isoprene, terpenes, alkanes, alkenes, carbonyls, alcohols, esters, ethers and acids. BVOC emissions are involved in stress resistance mechanisms (due to heat, water shortage, oxidation, herbivore or pathogen attack) and communication (plant-plant and plant-insect interactions) (Kesselmeier and Staudt, 1999). Emission rates depend on abiotic factors such as temperature and light, and biotic factors such as tree species, leaf

age and stress level (Niinemets et al., 2004; Loreto and Schnitzler, 2010; Niinemets, 2010). BVOC emissions are therefore highly variable in space and time. Unlike specific Anthropogenic Volatile Organic Compounds (AVOCs) such as benzene, emitted BVOCs may not be directly harmful to human health. However, BVOCs may form secondary pollutants, such as ozone (Calfapietra, 2013; Atkinson and Arey, 2003a; Churkina et al., 2017) and secondary organic aerosols (Salvador et al., 2020; C. Minguillón et al., 2016; Churkina et al., 2017; Lehtipalo et al., 2018). BVOCs emitted in the gaseous phase are oxidized in

the atmosphere, forming more functionalized compounds that are semi-volatile and may be absorbed into aerosols. In the urban VOC-limited environment with high nitrogen oxides (NOx) (emitted by traffic), ozone formation strongly depends on the VOC concentrations. There are also feedbacks between the urban environment, which is stressful for trees (higher temperatures and concentrations of oxidizing pollutants, difficult access to water) (Lüttge and Buckeridge, 2023; Czaja et al., 2020) and BVOC emissions.

To understand processes and forecast the evolution of pollutant concentrations, numerical models are widely used. Air-quality models of various types and resolutions exist, depending on the scale and processes studied. Chemistry-Transport Models (CTM) are eulerian models that represent the chemistry and transport of pollutants in three-dimensional grid cells, e.g. CHIMERE (Menut et al., 2021), Polair3D (Boutahar et al., 2004), WRF-Chem (NOAA/ESRL, 2023), CMAQ (Byun and



Schere, 2006; Appel et al., 2021), MOCAGE (CNRM, 2023), etc. Their typical horizontal resolution varies between 1 and
$10^2$ km and they are used from the global to the regional scales (Mailler et al., 2017). Many input data are necessary: surface
characterization, spatio-temporal emissions of each pollutant, boundary and initial conditions. Due to the coarse resolution, the
surface type is characterized by land-use categories such as open water, urban, forest, crop, etc. Forest trees are usually divided
into 1 to 5 categories based on general characteristics (evergreen or deciduous, broadleaf or needleleaf). Most of the CTMs
compute the BVOC emissions from forest and crops based on Plant Functional Type (PFT) and the MEGAN empirical model
(Model of Emissions of Gases and Aerosols from Nature) (Guenther et al., 2006, 2012; Matthias et al., 2018). The heterogene-
ity of the vegetation species is not explicitly modeled, but the model gives a rough estimate of the BVOC emission rates in the
grid cells containing vegetation. CTMs can be used to compute air quality over large urban areas, but at this spatial resolution
the land-use is urban and no biogenic emission from urban vegetation is taken into account. In parallel, tree inventories are
being developed in many cities (Bennett, 2023) and give us a much more accurate description of the urban forest. They can
contain the tree precise locations, species and sizes, allowing to study the impact of urban vegetation on air quality (Mircea
et al., 2023).

Based on the tree inventory implemented in Paris by the Municipality (Municipality of Paris, 2023) and a series of allomet-
ric equations (McPherson et al., 2016), a method is developed to estimate the BVOC emissions by urban trees in Paris. This
"bottom-up" inventory of BVOC emissions by urban trees is used to estimate emissions from Paris trees over June and July
2022. This period is chosen because biogenic emissions are expected to be higher in summer, especially during heatwaves,
and also because of the numerous in situ measurements that have been performed in Paris region. The effect on isoprene,
monoterpene, $O_3$ and particle concentrations over the Île-de-France (IDF) region is quantified using the CTM CHIMERE.

## 2 Materials and Methods

### 2.1 Tree-based BVOC emission model

To compute BVOC emissions in CTMs, an empirical approach is usually used. The emission rate of each BVOC species is
computed as the product of several factors: the amount of vegetation (surface of the land-use category, leaf area index or mass),
an emission factor at the standard conditions (leaf temperature of 30 °C and Photosynthetic Photon Flux Density (PPFD) of
1000 μmol photons.m$^{-2}$.s$^{-1}$) and activity factors representing physiological or meteorological effects. One emission factor
is associated to each PFT. The development of BVOC rate measurements at the leaf or branch scale with chambers and tree
75 inventories allows estimation of BVOC emission rates at the tree level (Owen et al., 2001, 2003; Stewart et al., 2003; Karl
et al., 2009; Steinbrecher et al., 2009). The emission rate of a BVOC $k$ for a tree $t$ (μg.h$^{-1}$ per tree) can be estimated as:

$$ER_{k,t} = DB_t \cdot EF_{k,t} \cdot \gamma_k \tag{1}$$

where:

- $EF_{k,t}$ (μg.g$_{DW}^{-1}$.h$^{-1}$) is the emission factor (or potential) at standard conditions,
80 - $DB_t$ is the dry leaf biomass (g$_{DW}$, where $DW$ stands for dry weight),
- $\gamma_k$ combines the different dimensionless emission activity factors.



## 2.2 Tree inventory and characteristics

The Paris Tree database (https://opendata.paris.fr/explore/dataset/les-arbres/map/) regroups an inventory of the public trees. Many mapped information is available for each tree: precise location (coordinates), address, type (roadside, garden, cemetery etc.), tree species, height, trunk circumference and development stage. It is regularly updated, and the version used in this study was downloaded in March 2023. A map of trees around Avenue des Champs-Élysées taken from the database is shown in Fig. B1. The proportion ($P$) of the tree genus found in Paris is presented in Fig. B2 and the distributions of their trunk circumferences and crown heights are shown in Fig. B3. The municipality of Paris estimates that around $1/3$ of the Parisian trees are missing from their database, mainly trees located in private areas. Without further information on these private trees, they are not taken into account in this study.

To compute the BVOC emissions (eq. (1)) of each individual tree, an estimation of the leaf dry biomass is necessary. Dry biomass such as leaf area, and crown dimensions can be estimated using allometric equations. These allometric relationships are statistical models based on a sample of measurements predicting tree size as a function of parameters such as trunk diameter or age since planting. Many studies propose equations for forest trees (Burton et al., 1991; Bartelink, 1997; Karlik and McKay, 2002), but studies on urban trees are more scarce (Nowak, 1996). The open database of McPherson et al. (2016) is chosen in this study because it was developed specifically for urban trees and includes many genus found in Paris (84% of the trees in the Paris inventory) (365 growth equations for 174 tree species). For missing tree genus, equation from another tree genus in the same family is selected, as described in Section 3. It includes allometric tree measurements for different climates in the United-States, so assumptions are necessary to select the climate for each tree species that is the closest to that of the Paris region (see Section 3).

## 2.3 Description of regional-scale air-quality simulations

To quantify the impact of the Parisian tree emissions on air quality, regional-scale simulations using the 3D CTM CHIMERE v2020_r3 (Menut et al., 2021) coupled to the chemical module SSH-aerosol v1.3 (Sartelet et al., 2020) are performed. The gas-phase chemical scheme is MELCHIOR2, modified to represent secondary organic aerosol formation, as described in Sartelet et al. (2020). Biogenic emissions are estimated using the MEGANv2.1 algorithm implemented in CHIMERE (Couvidat et al., 2018), which corresponds to a land-use approach. The following section describes the simulation setup.

Simulations are performed during summer 2022, between 6 June and 31 July 2022, with a five-day spin-up period (1-5 June). Summer time is chosen as biogenic emissions are the highest during this period due to meteorological conditions. In France, the summer 2022 was exceptionally hot and sunny, with little precipitation (on average 1 to 3 °C above seasonal values over most of France) (Meteo France, 2023). The domain of study corresponds to the Île-de-France region, with a 1 km × 1 km spatial resolution (IDF1). Initial and boundary conditions are taken from two additional nesting simulations: one over France with a 9 km × 9 km spatial resolution (FRA9), and one over the northwest of France with a 3 km × 3 km spatial resolution



(IDF3), as shown in Fig. 1. For the FRA9 domain, boundary and initial conditions are obtained from the CAMS platform (Inness et al., 2019), with a 10 km × 10 km spatial resolution.

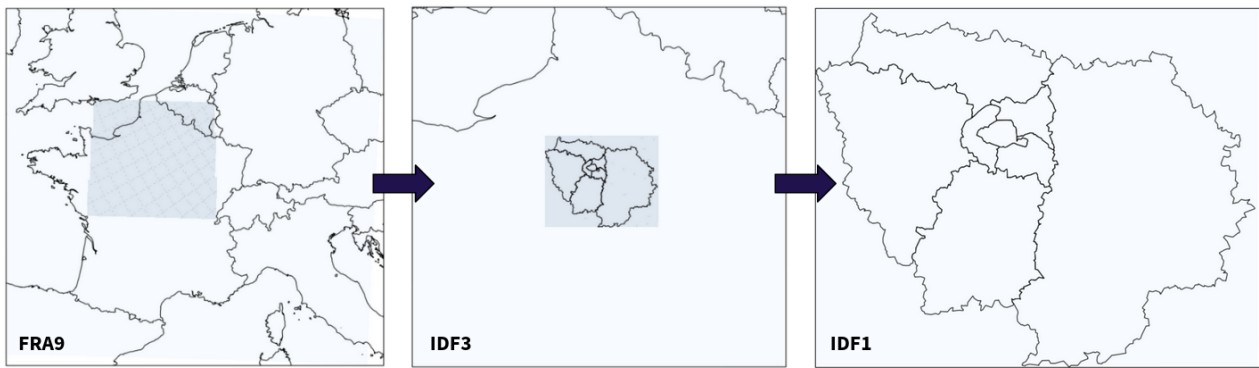

**Figure 1.** Representation of the simulated domains. The blue rectangles represent the location of the different nested domains.

Meteorological data for all domains are computed using the Weather Research and Forecasting (WRF) model v3.7.1 available in CHIMERE (Powers et al., 2017; Menut et al., 2021). Even if CHIMERE and WRF simulations are performed simultaneously, one-way coupling is used and then concentrations computed in CHIMERE are assumed to have no influence on the meteorological fields computed by WRF. WRF simulations are performed with 33 vertical levels, from 0 to 20 km altitude. A more refined vertical discretization is employed in the first four vertical levels (average heights of 0.12, 25, 50 and 83 meters, respectively), which contains almost all buildings in Paris region. In order to represent more precisely the meteorological fields in urban areas, the single-layer urban canopy model (UCM) (Kusaka et al., 2001) is used in the IDF3 and IDF1 domains. Three urban categories are employed to differentiate street and building dimensions, as well as heat transfer parameters in commercial, high and low intensity residential areas. The spatial distribution of each urban category used in WRF simulations is based on CORINE land-use coverage, with a 250 m resolution (available in https://doi.org/10.2909/71c95a07-e296-44fc-b22b-415f42acfdf0). In the single-layer UCM model, the sensible heat flux (AH) is assumed to be 45 $W.m^{-2}$ for commercial areas, 10 $W.m^{-2}$ for high intensity residential and 5 $W.m^{-2}$ for low intensity residential areas, based on Pigeon et al. (2007b) and Sailor et al. (2015). Table 1 summarizes the other physical options employed in the WRF simulations.

Anthropogenic emissions in the domains FRA9 and IDF3 are from the latest (2020) EMEP emission inventory (EMEP, 2019) (0.1°×0.1° horizontal resolution), and in IDF1, they are from the latest (2019) regional emission inventory of the Air Quality Monitoring Network (AQMN) Airparif for the Greater Paris area. (https://www.airparif.asso.fr/) (1 km × 1 km spatial resolution). Traffic emissions correspond to those of the summer 2022 calculated using the bottom-up traffic emissions model HEAVEN (https://www.airparif.asso.fr/heaven-emissions-du-trafic-en-temps-reel), while non-traffic anthropogenic emissions correspond to the 2019 Airparif inventory.





**Table 1.** Main physical options employed in WRF simulations.

| option in WRF namelist | option complete name | option selected |
| --- | --- | --- |
| mp_physics | Microphysics | Thompson graupel scheme |
| cu_physics | Cumulus | Grell-Devenyi ensemble scheme |
| ra_lw_physics | Longwave radiation | rrtmg scheme |
| ra_sw_physics | Shortwave radiation | rrtmg scheme |
| bl_pbl_physics | Boundary layer | YSU scheme |
| sf_sfclay_physics | Surface layer | Monin-Obukhov Similarity scheme |
| sf_surface_physics | Land surface | Noah Land-Surface Model |
| sf_urban_physics | Urban canopy model | Single-layer (only in IDF3 and IDF1) |

## 2.4 Description of the air-quality experimental measurements

The results of the simulations are compared to experimental measurements performed at different sites in the Paris region. In sections 4.1.2 and 4.2, temporal variations of observed and simulated concentrations are presented in three main sites: the Halles site, a permanent air-quality monitoring station located in the city center and operated by Airparif, the PRG-Paris Rive Gauche site, located at the 7[th] floor of the Lamark B building of Université Paris Cité (30 m above ground layer), in the south-east side of the city, set up as part of the ACROSS campaign (Cantrell and Michoud, 2022), and the SIRTA site (Site Instrumental de Recherche par Télédétection Atmosphérique), an atmospheric observatory located 20 km south-west of Paris which is integrated in the ACTRIS European Research Infrastructure Consortium (https://www.actris.eu) (Haeffelin et al., 2005). The Halles and PRG stations are both urban background sites, while SIRTA is a suburban background site. The three sites and the measurements performed are described in Table 2 and a more complete description of the measurements and their associated uncertainties is provided in Appendix A.

The reference simulations (without urban trees biogenic emissions) are validated in Section 4.1 with the observation sites of the Airparif network. These sites correspond to 21 permanent air-quality monitoring stations included within a large operational stations network operated by Airparif (see Table A1 and https://www.airparif.asso.fr/carte-des-stations). The map in Fig. 2 shows the location of all measurement stations which are used to evaluate the simulations (Sections 4.1 and 4.2). It also presents the land-use from GLOBCOVER (Team et al., 2011) used in the CHIMERE simulation over IDF1. It is mainly composed of agricultural lands, forests of varying sizes and a large urban area including Paris and its suburbs.



**Table 2.** Description of the experimental measurements performed at different sites and used in this study. ACSM: Aerosol Chemical Speciation Monitor (Petit et al., 2015), PTR-MS: Proton-Transfer-Reaction Mass Spectrometry (Simon et al., 2023). GC-FID: Gas-Chromatograph with a Flame Ionisation Detector (Gros et al., 2011)

| Site | location | typology | species measured | instrument |
|------|----------|----------|------------------|------------|
| Halles | 1$^{st}$ district of Paris city (48.862128° N, 2.344622° E) | urban background | $NO_2$ $O_3$ PM OM | AC32M O3 42e FIDAS 200 ACSM |
| PRG | 13$^{th}$ district of Paris city (48.827778° N, 2.380562° E) | urban background | $C_5H_8$ monoterpenes OM | PTR-MS PTR-MS ACSM |
| SIRTA | 20 km south-west of Paris (48.709890° N, 2.147938° E) | suburban | $C_5H_8$ monoterpenes OM | GC-FID PTR-MS ACSM |

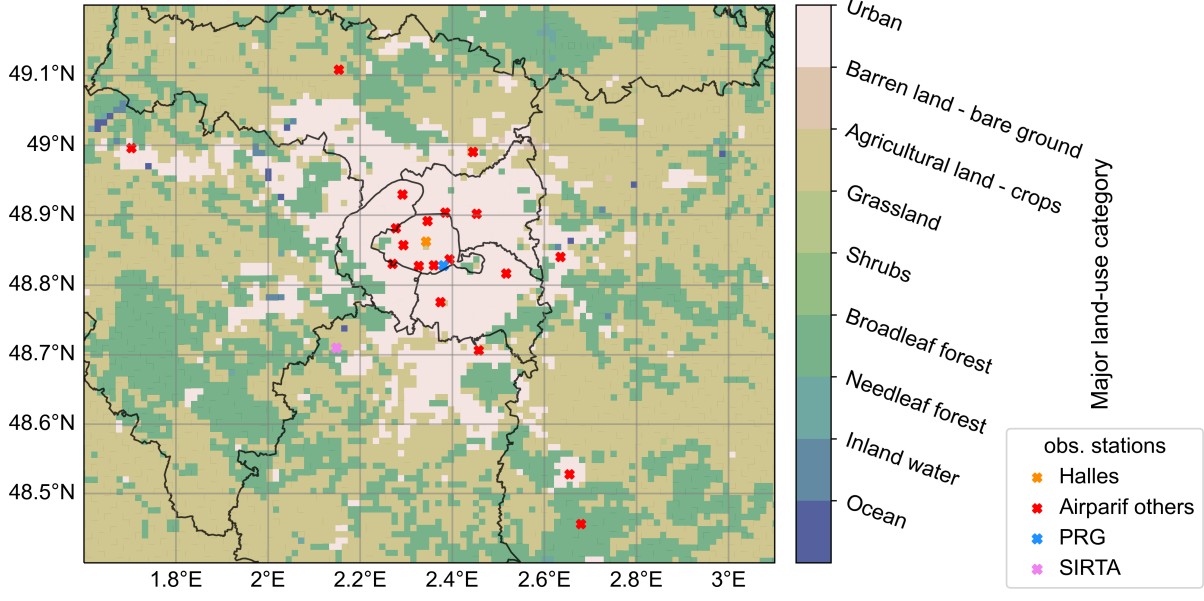

**Figure 2.** Map of the GLOBCOVER major land-use in each grid cell used in IDF1 CHIMERE simulations. The crosses represent the locations of the different measurement stations.



## 3 Bottom-up inventory of tree BVOC emissions and comparison to the land-use approach

### 3.1 Calculation of BVOC emissions at the tree level

#### 3.1.1 Estimation of the tree dry biomass

The total tree leaf dry biomass (in grams of dry weight, $g_{DW}$) is computed based on the McPherson et al. (2016) allometric equation database. Tree data were collected in 17 reference cities representative of the different US climate zones and analyzed to obtain growth equations. The database contains equations to estimate the tree characteristics from the tree species, climate and the trunk diameter at breast height (at 1.3 m) (DBH). To find the correspondence between Parisian trees and this database, the US climates were first ranked from closest to farthest from the Parisian climate based on a qualitative comparison of annual

rainfall and temperatures (see Table 3).

**Table 3.** US reference cities and climates used in the McPherson et al. (2016) study ranked from the closest to the farthest from the Parisian climate. The last column refers to Köppen climate classification (Paris region: Cfb).

| Rank | Region Code | Region Name | City | State | Climate class |
|------|-------------|-------------|------|-------|---------------|
| 1 | NoEast | Northeast | Queens | New York | Cfa |
| 2 | Piedmt | South | Charlotte | North Carolina | Cfa |
| 3 | LoMidW | Lower Midwest | Indianapolis | Indiana | Cfa |
| 4 | GulfCo | Coastal Plain | Charleston | South Carolina | Cfa |
| 5 | CenFla | Central Florida | Orlando | Florida | Cfa |
| 6 | PacfNW | Pacific Northwest | Longview | Oregon | Csb |
| 7 | TpIntW | Temperate Interior West | Boise | Idaho | Csa |
| 8 | NoCalC | Northern California Coast | Berkeley | California | Csb |
| 9 | InlEmp | Inland Empire | Claremont | California | Csb |
| 10 | SoCalC | Southern California Coast | Santa Monica | California | Csb |
| 11 | SacVal | Sacramento Valley | Sacramento | California | Csa |
| 12 | NMtnPr | North | Fort Collins | Colorado | Dfb |
| 13 | InterW | Interior West | Albuquerque | New Mexico | Bsk |
| 14 | MidWst | Midwest | Minneapolis | Minnesota | Dfa |
| 15 | InlVal | Inland Valleys | Modesto | California | Bsk |
| 16 | SWDsrt | Southwest Desert | Glendale | Arizona | Bwh |
| 17 | Tropic | Tropical | Honolulu | Hawaii | As |





For each tree species in the Paris tree inventory, the allometric equations are obtained from the US database, by selecting the closest tree category in terms of tree species and climate following the decision tree shown in Fig. 3. The default species is the plane tree (*Planatus* x *hispanica*), which is the predominant species in Paris.

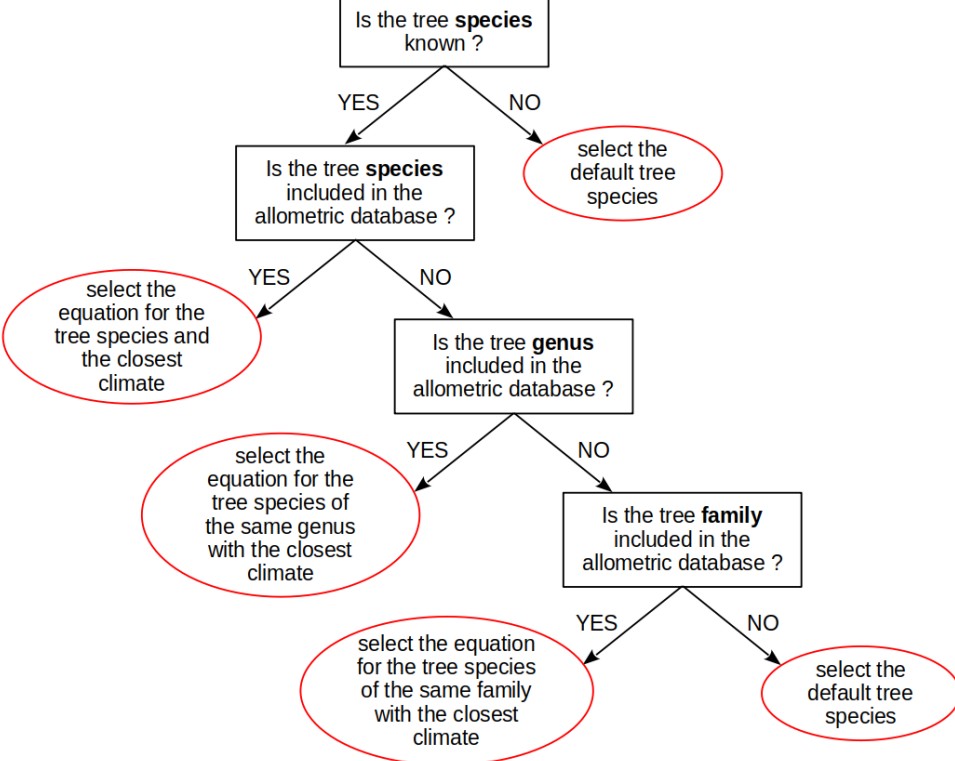

**Figure 3.** Decision tree to select the tree category to be used for each Paris tree. The tree category and corresponding allometric database refers to McPherson et al. (2016).

Then, the trunk diameter at breast height, $DBH$ (cm), is computed from the trunk circumference, $CIRC$ (cm) available in
the Paris tree inventory for each tree, assuming a cylindrical tree trunk, as:

$$DBH = \frac{CIRC}{\pi}. \tag{2}$$

The total tree leaf area ($LA$ in m$^2$) is then computed from each Parisian tree using the selected equation form and coefficient and the computed $DBH$. For example, the function $LA = f(DBH)$ is shown for three tree species in equations (3), (4) and (5), where $a$, $b$, $c$, $d$ and $MSE$ (Mean Squared Error) are dimensionless model coefficients.

For *Planatus* x *hispanica* (London plane):

$$LA = \exp\left[a + b\ln\left(\ln\left(DBH + 1\right)\right) + \frac{MSE}{2}\right],$$

with $a = -2.06877$, $b = 5.77886$ and $MSE = 0.27978$. $\qquad\qquad$ (3)



For *Acer platanoides* (Norway maple):

$$LA = \exp\left[a + b\ln\left(\ln\left(DBH + 1\right)\right) + \left(\sqrt{DBH} \times \frac{MSE}{2}\right)\right],$$

with $a = -0.55184$, $b = 4.27852$ and $MSE = 0.07518$. \hfill (4)

For *Prunus serrulata* (Japanese cherry):

$$LA = a + b\,DBH + c\,DBH^2 + d\,DBH^3,$$

with $a = -18.045$, $b = 4.6553$ $c = -0.12798$ and $d = 0.00198$. \hfill (5)

Finally, the dry biomass ($DB$ in $g_{DW}$) is the product of the leaf area and the dry weight density ($DWD$ in $g_{DW}.m^{-2}$):

$$DB_t = LA_t \times DWD_t. \tag{6}$$

The dry weight density depends on tree species and it is also given in the McPherson et al. (2016) database. For instance, $DWD = 500$, $520$ and $560\ g_{DW}.m^{-2}$ for *Planatus* x *hispanica*, *Acer platanoides* and *Prunus serrulata* respectively.

The computed $LA$ and $DB$ are shown in Fig. 4 for the predominant tree species ($P > 1\%$) as a function of the $DBH$. It shows that $LA$ and $DB$ increase with $DBH$ but there is a large variability depending on the tree species. For example, for a tree of $DBH = 100$ cm, the estimation of the leaf surface is equal to $1151.5$ m$^2$ for *Planatus* x *hispanica*, $582.5$ m$^2$ for *Acer platanoides* and $1147.7$ m$^2$ for *Prunus serrulata*.

As the simulation is performed during the late spring and summer periods, the tree foliage is assumed to be fully developed,
so that the leaf area and dry biomass are constant over time. For longer simulations periods, the temporal evolution of leaf area and dry biomass should be introduced.




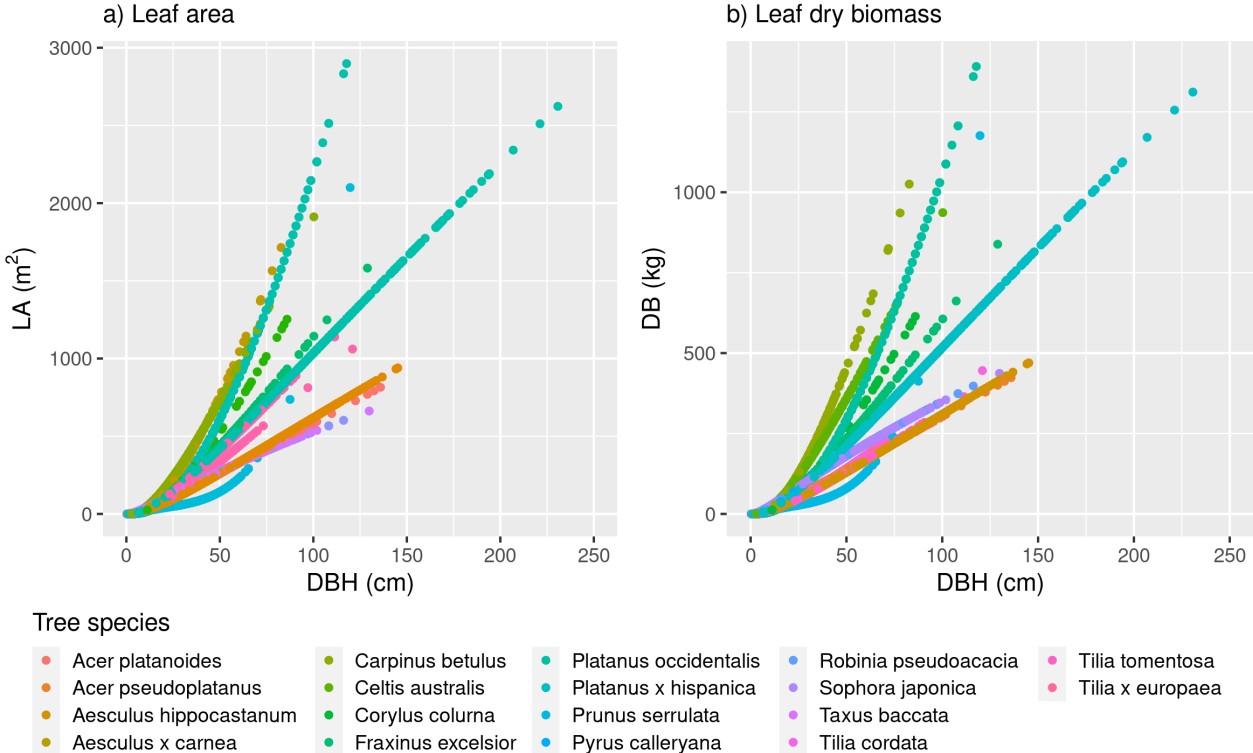

**Figure 4.** a) Leaf area ($LA$) and (b) dry biomass ($DB$) computed for the predominant tree species ($P > 1\%$) in the Paris city inventory as a function of $DBH$.

### 3.1.2 Emission factors by tree species

The emission factors by tree species are taken from MEGANv3.2 code, downloaded at https://bai.ess.uci.edu/megan/data-and-code/megan32, last accessed on 10/07/2023. The $EF$ presented by tree species are assumed to be identical within the same tree genus. Therefore, $EF$ by tree genus are used for all trees except for the Quercus genus (oak), whose species are known to have very different BVOC emission profiles (Loreto, 2002). The $EF$ for the Quercus species are taken from Ciccioli et al. (2023) for isoprene and monoterpenes. For the Quercus species missing in Ciccioli et al. (2023) but with a known emission type (Loreto, 2002), the $EF$ values are taken from MEGANv3.2. For unknown emission type, the $EF$ value is set by default to oak isoprene emitters in MEGANv3.2. For all tree species the $EF$ values for sesquiterpenes and oxygenated VOC are taken from MEGANv3.2. The emission factors of nitric oxide (NO) and carbon monoxide (CO) are fixed for all tree species and are equal to: $EF_{NO} = 0.05$ and $EF_{CO} = 1.0\,\mu\mathrm{g.g}_{DW}^{-1}.\mathrm{h}^{-1}$, as suggested in MEGANv3.2. The emission factors of isoprene (ISOP), total monoterpenes (MT), total sesquiterpenes (SQT) and total other VOCs (OVOC) are shown in Table 4 for the predominant tree genus and oak species.





**Table 4.** Emission factors ($EF$ in µg.g$_{DW}^{-1}$.h$^{-1}$) of BVOCs for the predominant tree genus found in Paris ($P > 1\%$) and for the predominant Quercus species. ISOP: isoprene, MT: monoterpenes, SQT: sesquiterpenes and OVOC: other VOC

| Genus | species | % of trees in Paris | $EF_{\text{ISOP}}$ | $EF_{\text{MT}}$ | $EF_{\text{SQT}}$ | $EF_{\text{OVOC}}$ |
|---|---|---|---|---|---|---|
| Platanus | all | 22.7 | 24 | 0.51 | 0.10 | 4.64 |
| Aesculus | all | 11.9 | 0 | 0.58 | 0.10 | 4.64 |
| Tilia | all | 10.3 | 0 | 0.53 | 0.10 | 4.64 |
| Acer | all | 7.7 | 0 | 0.51 | 0.10 | 4.64 |
| Sophora | all | 6.3 | 5.0 | 0.53 | 0.10 | 4.64 |
| Prunus | all | 3.9 | 0 | 1.18 | 0.10 | 4.64 |
| Fraxinus | all | 2.6 | 0 | 0.26 | 0.10 | 4.64 |
| Pyrus | all | 2.6 | 0 | 0.68 | 0.10 | 4.64 |
| Celtis | all | 2.3 | 0 | 0.33 | 0.10 | 4.64 |
| Pinus | all | 2.2 | 0 | 1.43 | 0.15 | 6.94 |
| Carpinus | all | 1.5 | 0 | 1.07 | 0.10 | 4.64 |
| Populus | all | 1.5 | 37 | 0.44 | 0.10 | 4.64 |
| Malus | all | 1.5 | 0 | 0.44 | 0.10 | 4.64 |
| Corylus | all | 1.4 | 1.0 | 1.81 | 0.10 | 4.64 |
| Robinia | all | 1.2 | 20 | 0.23 | 0.10 | 4.64 |
| Ulmus | all | 1.1 | 0 | 0.62 | 0.10 | 4.64 |
| Taxus | all | 1.1 | 0 | 0.58 | 0.15 | 4.64 |
| Betula | all | 1.0 | 0 | 0.66 | 0.10 | 4.64 |
| Gleditsia | all | 1.0 | 0 | 0.56 | 0.10 | 4.64 |
| Quercus | ilex | 0.485 | 0.1 | 43 | 0.10 | 4.64 |
| Quercus | robur | 0.365 | 70 | 0.3 | 0.10 | 4.64 |
| Quercus | rubra | 0.272 | 35 | 0.1 | 0.10 | 4.64 |
| Quercus | cerris | 0.257 | 0.1 | 0.6 | 0.10 | 4.64 |
| Quercus | petraea | 0.045 | 45 | 0.3 | 0.10 | 4.64 |
| Quercus | pubescens | 0.044 | 70 | 0.3 | 0.10 | 4.64 |
| Quercus | frainetto | 0.036 | 85 | 0.0 | 0.10 | 4.64 |
| Quercus | palustris | 0.035 | 34 | 1.0 | 0.10 | 4.64 |
| Quercus | coccinea | 0.025 | 34 | 1.0 | 0.10 | 4.64 |
| Quercus | suber | 0.018 | 0.2 | 20 | 0.10 | 4.64 |
| Quercus | coccifera | 0.016 | 0.1 | 25 | 0.10 | 4.64 |
| Quercus | phellos | 0.013 | 34 | 1.0 | 0.10 | 4.64 |
| Quercus | imbricaria | 0.011 | 34 | 1.0 | 0.10 | 4.64 |



### 3.1.3   Choice of activity factors

Emission factors, which are measured at standard conditions, are then multiplied by dimensionless factors representing variations of emissions as a function of biotic and abiotic processes. Photosynthetic Photon Flux Density (PPFD) is the flux of photons in the 400-700 nm spectral range of solar radiation that is used for photosynthesis. It is expressed in µmol.m$^{-2}$.s$^{-1}$ and is calculated from the simulated solar radiation in the grid cell where the tree is located $(i,j)$ as:

$$PPFD_{t\in(i,j)} = 4.5 \times 0.5 \times SWg_{t\in(i,j)}, \tag{7}$$

where $SWg$ is the global solar radiation (short wave), $4.5$ is a factor to convert the W.m$^{-2}$ into µmol.m$^{-2}$.s$^{-1}$ and $0.5$ is an approximation of the fraction of the solar radiation energy that is in the 400-700 nm spectral range (Meek et al., 1984).

For each BVOC, the activity factors for light (PPFD), $\gamma_{P_k,t\in(i,j)}$, and for temperature, $\gamma_{T_k,t\in(i,j)}$, are computed as the weighted average of a light-dependent ($LDF_k$) and light-independent fraction ($LIF_k = 1 - LDF_k$):

$$\gamma_{P_k,t\in(i,j)} = (1 - LDF_k) + LDF_k\,\gamma_{P\_LDF_k,t\in(i,j)} \tag{8}$$

$$\gamma_{T_k,t\in(i,j)} = (1 - LDF_k)\,\gamma_{T\_LIF_k,t\in(i,j)} + LDF_k\,\gamma_{T\_LDF_k,t\in(i,j)}. \tag{9}$$

The $LDF_k$ factor depends on the BVOC compound and can be found in Table 4 of Guenther et al. (2012).

**- Light effect $\gamma_{P_k,t\in(i,j)}$**

As no canopy model is used to consider the shadow effects inside the canopy, no distinction between the sunlit and shaded
leaves can be done. Therefore, the dependency to the past PPFD that require this distinction is not included and, the light activity factor is computed as (Guenther et al., 1995):

$$\gamma_{P\_LDF_k,t\in(i,j)} = \frac{C_P\,\alpha\,PPFD_{t\in(i,j)}}{\sqrt{1 + \alpha^2\,PPFD^2_{t\in(i,j)}}}, \tag{10}$$

with $\alpha = 0.004$ and $C_P = 1.03$.

**- Temperature effect $\gamma_{T_k,t\in(i,j)}$**

$$\gamma_{T\_LDF_k,t\in(i,j)} = \frac{E_{opt_k,t\in(i,j)}\,C_{T2}\,\exp\left(C_{T1_k}\,x_{t\in(i,j)}\right)}{C_{T2} - \left(C_{T1_k}\left[1 - \exp\left(C_{T2}\,x_{t\in(i,j)}\right)\right]\right)} \tag{11}$$

$$\gamma_{T\_LIF_k,t\in(i,j)} = \exp\left[\beta_k\left(T_{t\in(i,j)} - T_s\right)\right], \tag{12}$$

$$\text{with } x_{t\in(i,j)} = \frac{1}{R}\left(\frac{1}{T_{opt_{t\in(i,j)}}} - \frac{1}{T_{t\in(i,j)}}\right), \tag{13}$$

and $R = 0.00831$, $T_{t\in(i,j)}$ is the leaf surface temperature (K), approximated here by the air temperature at 2 m above ground layer in the horizontal grid cell $(i,j)$ to which the tree $t$ belongs.





$C_{T2} = 230, T_{opt_{t \in (i,j)}}, E_{opt_{k,t \in (i,j)}}$ are empirical coefficients:

$$T_{opt_{t \in (i,j)}} = 313 + 0.6 \left( T_{240_{t \in (i,j)}} - T_s \right) \tag{14}$$

$$E_{opt_{k,t \in (i,j)}} = C_{eo_k} \exp \left[ 0.05 \left( T_{24_{t \in (i,j)}} - T_s \right) \right] \exp \left[ 0.05 \left( T_{240_{t \in (i,j)}} - T_s \right) \right] \tag{15}$$

where $T_{24_{t \in (i,j)}}$ and $T_{240_{t \in (i,j)}}$ are the temperature averages over the past 24 and 240 h and $T_s = 297$ K is the standard conditions for leaf temperature. $C_{T1_k}$, $C_{eo,k}$ and $\beta_k$ are BVOC dependent empirical coefficients that can be found in the Table 4 of Guenther et al. (2012). To illustrate the variation of the activity factors with light and temperature, Fig. 5 shows averaged $\gamma_T$ and $\gamma_P$ for isoprene, $\alpha$-pinene and $\beta$-pinene. Figure 5a shows that BVOC emissions increase with temperature. At high temperatures, isoprene emissions are capped, while monoterpene emissions rise sharply. BVOC emissions also increase with

light (Fig. 5b) and the activity factor reaches its maximum value of $1$ after $PPFD = 1000$ μmol photons.m$^{-2}$.s$^{-1}$.

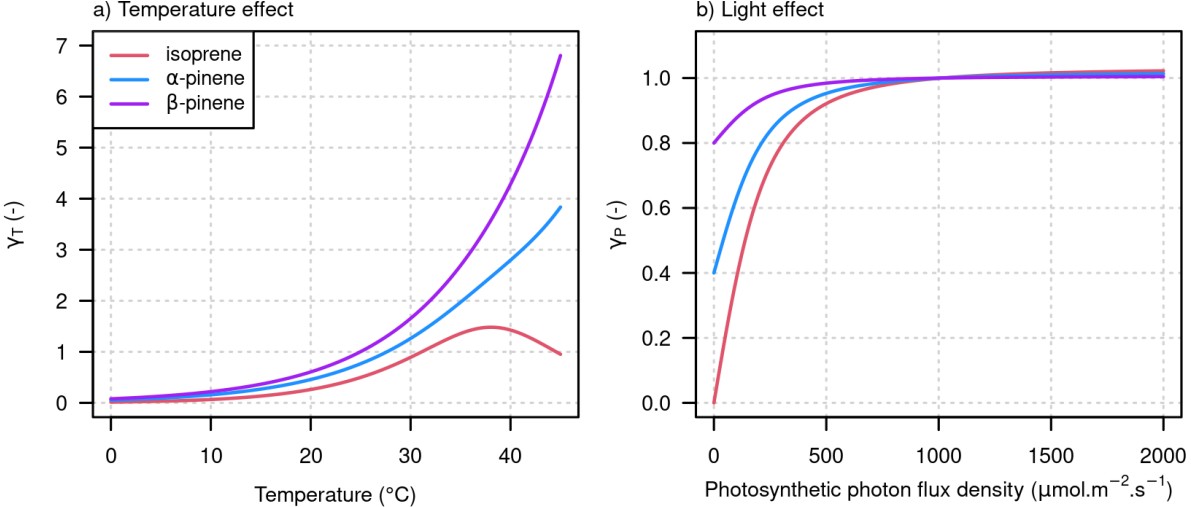

**Figure 5.** Dependence of activity factors on (a) temperature and (b) light variations for three BVOCs ($T_{24}$ and $T_{240}$ are fixed to 294 K in this figure) (Guenther et al., 2012).

**- Other factors**

Other activity factors could be added to represent the effect of leaf age and water stress. In this study, emissions are calculated per amount of leaf biomass, considering an average emission for all leaves in the canopy. In addition, we assume that in June and July, tree foliage is fully developed and leaf area and dry biomass are constant. Therefore, no activity factor is added to

modulate emissions according to the fraction of growing, mature, growing and old foliage (Guenther et al., 2012).

Several studies also introduce an activity factor to represent the impact of soil moisture and water stress on isoprene emissions (Guenther et al., 2012; Jiang et al., 2018; Bonn et al., 2019; Otu-Larbi et al., 2020; Wang et al., 2022). Although urban trees planted in reduced soil volumes may be subject to water stress (Lüttge and Buckeridge, 2023), the resolution of the CTM does not allow us to accurately simulate the soil water content in an urban environment, so no activity factor modulating isoprene

emissions as a function of water content is taken into account here.



## 3.2 Integration of individual tree BVOC emissions in CHIMERE

After estimating the biogenic emissions of each tree in the city of Paris, these emissions are integrated into the CHIMERE CTM. To do this, they must be spatialized and speciated, as detailed in this section.

### 3.2.1 Integration of individual tree BVOC emissions on the CTM grid

First, each tree is located within the CTM grid using its precise position given in the Paris Tree inventory and the coordinates of the CTM grid. The product of the dry biomass and the emission factor ($DB_t\,EF_{t,k}$) is then summed for all trees belonging to the same cell to compute the BVOC emission rates ($ER_{i,j,k}$ in µg.m$^{-2}$.h$^{-1}$) as:

$$ER_{i,j,k} = \frac{1}{\Delta x_{i,j}\Delta y_{i,j}} \sum_{t \in i,j} (DB_t\,EF_{t,k})\ \gamma_{T_{i,j,k}} \cdot \gamma_{P_{i,j,k}}, \tag{16}$$

where $\Delta x_{i,j}$ and $\Delta y_{i,j}$ are the size of the cell $(i,j)$ in the $x$ and $y$ directions (m), here both equal to 1000 m. A map of the
dry biomass integrated on the CTM grid cells is shown in Fig. 6. The average cell dry biomass over Paris is 130 g.m$^{-2}$ and can reach 390 g.m$^{-2}$ in cells containing large parks or cemeteries. The Paris tree inventory does not include all the trees in the Vincennes and Boulogne woods, however, these large woods are considered at the regional scale, so their emissions are calculated using the land-use approach as shown in Fig. 7b.

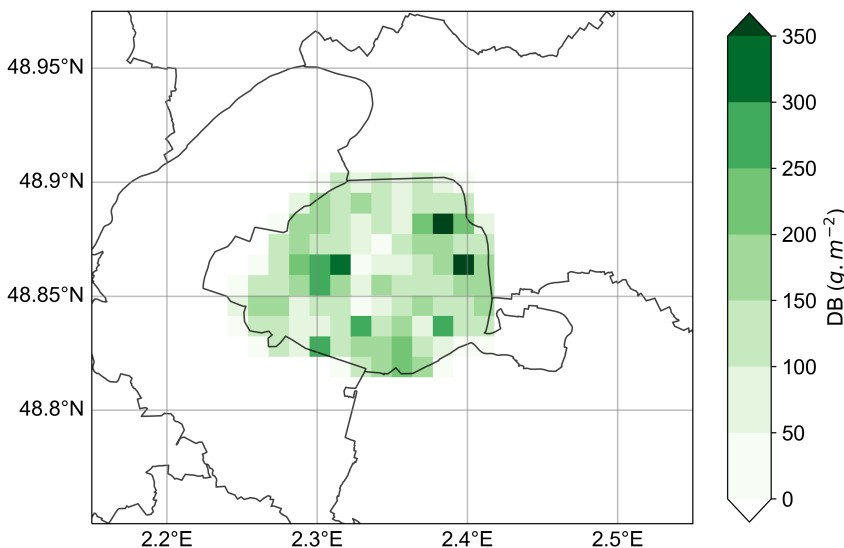

**Figure 6.** Tree leaf dry biomass computed over Paris from the Paris tree database and McPherson et al. (2016) with a spatial resolution of 1 km × 1 km.





### 3.2.2 Speciation and aggregation of BVOC species

The emission factors of MEGANv3.2 are estimated for different categories of BVOCs, which are presented in the rows of Table C1. These BVOC categories need to be disaggregated into model species to be used in the CTM. The chemical scheme used in CHIMERE corresponds to MELCHIOR2, and the model species are shown in the columns of Table C1. To disaggregate the BVOC categories into model species, the BVOC categories are first speciated into detailed real species, which are then aggregated into the model species. The speciation in real BVOC species is done with a speciation matrix available in

MEGANv3.2 code (downloaded at https://bai.ess.uci.edu/megan/data-and-code/megan32). Then, the real BVOC species used in CHIMERE are speciated and aggregated into MELCHIOR2 species. The product of the two matrices gives the speciation/aggregation matrix, described in Table C1. Note that no specific speciation is applied to sesquiterpenes, which are all included in the model species humulene (HUMULE). Monoterpenes are speciated as $\alpha$-pinene (APINEN), $\beta$-pinene (BPINEN), limonene (LIMONE) and ocimene (OCIMEN); and other VOCs (OVOC) represents ethylene (C2H4) and oxygenated VOCs (CH3OH,

CH3CHO, CH3COE and MEMALD).

Then, the BVOC emissions from urban trees in Paris are added to the regional-scale BVOC emissions to estimate the BVOC emissions over the Île-de-France region. The Section below details the complementarity between the bottom-up inventory for urban trees and the regional-scale PFT-based emissions.

### 3.3 Complementarity of the emissions computed by the bottom-up and the land-use approaches

At the regional-scale, biogenic emissions are estimated using a land-use approach with emission factors that depend on the land-use and PFT, as described in Guenther et al. (2012). As the land-use is urban over Paris, vegetation is not considered, and there are no biogenic emissions, as shown in Fig. 7, which represents on the left panel (a) the 2-month averaged isoprene emissions computed with the bottom-up inventory and on the right panel (b) isoprene emissions computed with the land-use approach in CHIMERE. The bottom-up inventory allows accounting of local biogenic emissions in the city but Fig. 7

shows that tree inventory and emissions are probably still missing in the Paris suburban area, because there is currently no tree inventory for most of the urban areas outside Paris city. The order of magnitude of isoprene emissions computed by the bottom-up inventory seems coherent compared to regional-scale emissions. Emission rates in Paris (0.12 µg.m$^{-2}$.s$^{-1}$ on average for isoprene) are lower than those simulated over the large Île-de-France forests. This is also the case for other BVOC species as shown in Appendix C. The relative distribution of monoterpenes emitted is different between the urban and the regional scales

as shown in Fig. C2. In particular, there is relatively more $\beta$-pinene in the regional-scale emissions. This is due to the different vegetation species between the city and the regional scale, and to the speciation of monoterpenes, which may be different.





**Figure 7.** Comparison of the 2-month averaged isoprene emissions with (a) the "bottom-up" inventory and (b) with the land-cover approach in CHIMERE over Île-de-France and Greater Paris.

The temporal variation of the spatially averaged emissions of different biogenic compounds is shown in Fig. 8. For all compounds, emissions are strongly correlated with temperature and sunlight. Over the 2-month periods, there are three emission peaks corresponding to periods of heatwaves with clear-sky conditions and air temperature reaching 35 °C. The impact of BVOC emissions on air quality is expected to be higher during these periods. Therefore, the effect of emissions on pollutant concentrations will be calculated both on the 2-month period and on the heatwave periods, which correspond to the following days: June, 15 to 18, July, 11 to 14 and 17 to 19. In terms of emitted compounds, isoprene is the most emitted biogenic species, followed by OVOC. Monoterpenes and CO are emitted to a lesser extent, followed by sesquiterpenes and NO. This distribution






of emissions is fairly typical of emissions calculated using the MEGAN model (Guenther et al., 2012; Ciccioli et al., 2023).
In terms of emission intensity, some recent studies computing the BVOC emissions over Europe with plant emission specific models instead of using the PFT approach of MEGAN, have reported that isoprene emissions may be overestimated by a factor 3 in MEGANv2.1, while monoterpene and sesquiterpene emissions may be underestimated by a factor 3 especially in summer (Jiang et al., 2019; Ciccioli et al., 2023). These discrepancies were attributed to the different vegetation classifications and emission factors at standard conditions. Using plant emission specific models, Jiang et al. (2019) found a better comparison to
observations for isoprene and organic aerosol concentrations at the European scale, while around the Paris basin in summer, differences in emissions mainly concern monoterpenes and sesquiterpenes. In order to take these emission uncertainties into account in our study, sensitivity simulations are carried out by multiplying monoterpene and sesquiterpene emissions by a factor 2 or 3.

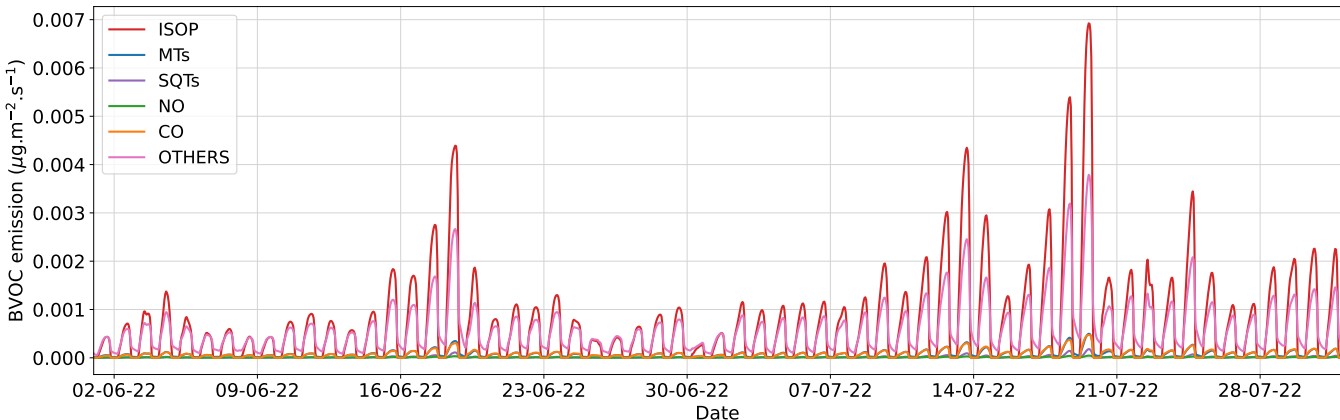

**Figure 8.** Temporal variation of the spatially averaged biogenic emissions computed over Paris with the bottom-up inventory.





## 4 Quantification of the impact of BVOC emissions from urban trees on air quality at the regional scale

Before studying the impacts of the bottom-up inventory, comparisons of simulated and observed key variables are performed to evaluate the simulation performance. Meteorological variables are first analyzed in Section 4.1.1, as biogenic emissions are strongly related to them. Then, Section 4.1.2 presents comparisons of modeled and observed pollutant concentrations at different air-quality stations in Île-de-France. Simulations are performed with the emissions factors presented above (REF) and with monoterpene and sesquiterpene emissions multiplied by a factor 2 (REF-TX2) and 3 (REF-TX3).

Then, to quantify the impacts of urban trees on air quality, simulations with the biogenic emissions from urban trees are performed and compared to the simulations without trees in Section 4.2. Three simulations with urban trees are performed: one for each monoterpene and sesquiterpene emissions scenario, which are referred to as bioparis, bioparis-TX2 and bioparis-TX3. All the simulations performed and the corresponding emissions are presented in Table 5.

**Table 5.** Simulation list with corresponding emission scenarios. $ER$ stands for emission rates, SQT for sesquiterpenes, MT for monoterpenes, LUA for land-use approach and BUI for bottom-up inventory.

| Simulation name | Emissions computed with the land-use approach over IDF | Emissions computed with the bottom-up inventory over Paris |
| --- | --- | --- |
| REF | yes | no |
| REF-TX2 | yes $ER^{\mathrm{LUA}}_{\mathrm{MT\,\&\,SQT}} \times 2$ | no |
| REF-TX3 | yes $ER^{\mathrm{LUA}}_{\mathrm{MT\,\&\,SQT}} \times 3$ | no |
| bioparis | yes | yes |
| bioparis-TX2 | yes $ER^{\mathrm{LUA}}_{\mathrm{MT\,\&\,SQT}} \times 2$ | yes $ER^{\mathrm{BUI}}_{\mathrm{MT\,\&\,SQT}} \times 2$ |
| bioparis-TX3 | yes $ER^{\mathrm{LUA}}_{\mathrm{MT\,\&\,SQT}} \times 3$ | yes $ER^{\mathrm{BUI}}_{\mathrm{MT\,\&\,SQT}} \times 3$ |

### 4.1 Validation of the reference simulations

### 4.1.1 Meteorology

The surface meteorological fields simulated by WRF-CHIMERE are compared to measurements performed at SIRTA. The 10-minute averages of meteorological measurements of air temperature (T), relative humidity (RH), pressure (P), precipitation at 2 m, wind speed and direction at 10 m above ground level, as well as longwave (LW), global shortwave (SW) and PPFD incident radiations at the surface are compared. The wind speed and direction observed at 10 m are approximated by the value

simulated in the grid cell from 0.15 to 24 m, that is supposed to represent the field at the mid-cell altitude (i.e. $\approx 12$ m). The meteorological fields are extracted in the horizontal cell of the IDF1 domain which includes the SIRTA. Figure 9 shows the comparison of modeled and observed air temperature at 2 m height and PPFD, which are the two meteorological variables used to calculate BVOC emissions, from June to July 2022. They are also compared with statistical indicators (defined in Appendix D) in the Table 6 along with the other simulated and observed meteorological variables.




Figure 9 and Table 6 show that the variations of air temperature at 2 m and PPFD are well modeled with high correlations and low errors. The temperature is slightly overestimated by the model, especially after the 16th of July, resulting in an average positive bias of about 1 °C. For PPFD, the daily maximum is overestimated some days resulting in a positive bias. As PPFD is computed from the global solar radiation (SW) and the bias on this global solar radiation is lower, this overestimation may also come from the conversion coefficient between PPFD and solar radiation. Some tests have been performed to compare the BVOC emission of the bottom-up inventory calculated with the PPFD/SW ratio measured at SIRTA instead of the ratio used by CHIMERE (2.25) and showed that the impact on BVOC emissions was not significant. Other meteorological variables such as air relative humidity, pressure, wind direction and incident longwave radiation are also well modeled (Table 6). The wind speed is slightly overestimated but this may be due to a difference in representativity between a punctual wind speed measurement and an average value in the 24m-thick vertical cell. The low rainfall intensity limits the significance of statistical indicator calculations but the temporal comparison (not shown) demonstrates that the rainy days are well represented by the model, but the intensity of heavy rains is underestimated.

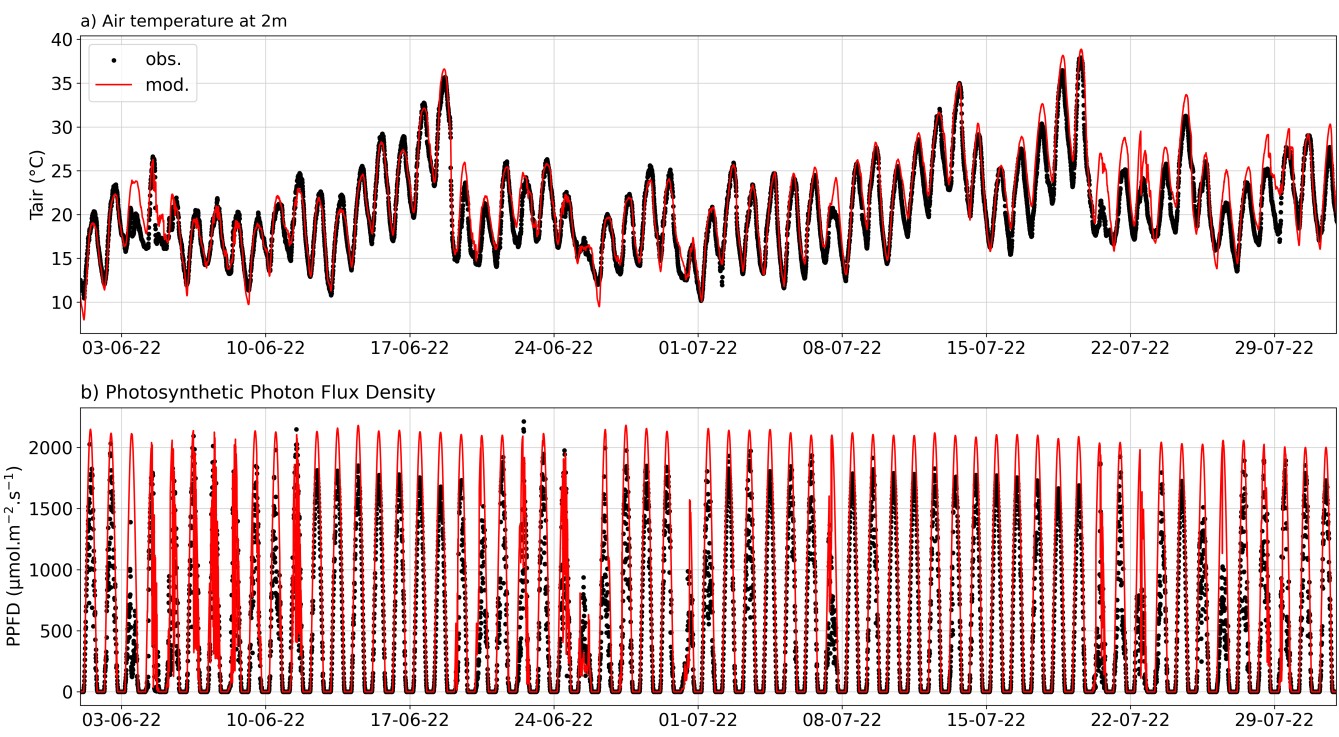

**Figure 9.** Comparison of the temporal variation of (a) air temperature at 2 m height and (b) photosynthetic photo flux density modeled by WRF (mod) and observed (obs) at the SIRTA observatory site (48.717347° N, 2.208868° E).





**Table 6.** Statistical indicators for the comparison of the meteorological variables simulated by WRF and observed at the SIRTA observatory site (48.7° N, 2.2° E). RMSE: Root Mean Square Error, NAD: Normalized Absolute Difference, Bias: Fractional Mean Bias and $R$: Pearson correlation coefficient (Appendix D), AGL: above ground layer, su: same unit as the meteorological variable.

| Obs. height | Variable | Unit | mean obs. | mean mod. | RMSE | NAD | Bias | $R$ |
| | | | su | su | su | - | su | - |
| --- | --- | --- | --- | --- | --- | --- | --- | --- |
| | T | °C | 20.5 | 21.4 | 1.80 | 0.03 | 0.94 | 0.96 |
| | RH | % | 57.4 | 51.1 | 11.23 | 0.08 | −6.27 | 0.88 |
| 2m AGL | P | hPa | 998.1 | 1000.8 | 2.82 | 0.00 | 2.72 | 0.99 |
| | rain | mm | 0.008 | 0.010 | 0.19 | 0.83 | 0.00 | 0.05 |
| | wind speed | m.s$^{-1}$ | 2.6 | 4.5 | 2.37 | 0.28 | 1.85 | 0.58 |
| 10m AGL | wind direction | ° | 179.2 | 179.2 | 88.26 | 0.14 | 1.63 | 0.68 |
| | PPFD | µmol.m$^{-2}$.s$^{-1}$ | 524.5 | 733.5 | 399.66 | 0.19 | 205.78 | 0.91 |
| surface | SW | W.m$^{-2}$ | 278.6 | 326.0 | 153.30 | 0.12 | 45.26 | 0.91 |
| | LW | W.m$^{-2}$ | 347.5 | 338.8 | 23.31 | 0.02 | −8.20 | 0.72 |

### 4.1.2 Model to data comparisons of gas and particle concentrations

In this section the $NO_2$, $O_3$, OM, $PM_{2.5}$, isoprene and monoterpene concentrations simulated by CHIMERE are compared to observations performed at different measurement stations over Île-de-France. The concentrations simulated in the horizontal grid cell containing the station and in the first vertical layer are compared to the observed concentrations in Table 7 for the three emission scenarios REF, REF-TX2 and REF-TX3. Two performance criteria are defined by Hanna and Chang (2012), and they are used here to evaluate the simulations performance. The most strict criteria are accepted when $-0.3 <$ FB $< 0.3$, $0.7 <$ MG $< 1.3$, NMSE $< 3$, VG $< 1.6$, FAC2 $\geq 0.5$, NAD $< 0.3$. The less strict criteria are accepted when $-0.67 <$ FB $< 0.67$, NMSE $< 6$, FAC2 $\geq 0.3$, NAD $< 0.5$ (where FB: Fractional Bias, MG: Geometric Mean Bias, NMSE: Normalized Mean Square Error, VG: Geometric Variance, FAC2: Factor of 2, NAD: Normalized Absolute Difference and $R$: correlation coefficient, see Appendix D). Values that respect the most strict performance criteria are represented in bold, those that respect the acceptable performance criteria for urban areas are represented in italics, and those that do not respect any criteria are in normal font. In order to investigate in more detail the model performance in each simulation, the temporal evolution of simulated and observed concentrations in three different stations (the Halles and PRG urban stations, and the SIRTA suburban station) is presented in Figures 10 and 11.



**Table 7.** Statistical comparison of the observed and simulated concentrations on average over 21 stations in IDF1 (listed in Table A1). Values indicated in **bold** respect the most strict performance criteria, while those in *italics* respect the acceptable performance criteria, and those in normal font do not respect the performance criteria defined by Hanna and Chang (2012). Correlation coefficients ($R$) are not included in the performance criteria. FB: Fractional Bias, MG: Geometric Mean Bias, NMSE: Normalized Mean Square Error, VG: Geometric Variance, FAC2: Factor of 2, NAD: Normalized Absolute Difference, $R$: correlation coefficient. The calculation of the statistical indicators and performance criteria are presented in Appendix D. su stands for same unit as the concentration.

| species & unit | simulation | Nb stat. | Obs. su | Sim. su | FB - | MG - | NMSE - | VG - | FAC2 - | NAD - | $R$ - |
|---|---|---|---|---|---|---|---|---|---|---|---|
| NO$_2$ µg.m$^{-3}$ | REF | 20 | 14.6 | 15.8 | **0.05** | **1.16** | **0.52** | **1.52** | **0.67** | **0.24** | 0.54 |
| | REF-TX2 | | | 15.7 | **0.04** | **1.15** | **0.52** | **1.52** | **0.67** | **0.24** | 0.54 |
| | REF-TX3 | | | 15.6 | **0.04** | **1.15** | **0.53** | **1.52** | **0.67** | **0.24** | 0.54 |
| O$_3$ µg.m$^{-3}$ | REF | 12 | 68.2 | 82.9 | **0.19** | **1.29** | **0.13** | **1.31** | **0.85** | **0.14** | 0.69 |
| | REF-TX2 | | | 83.8 | **0.21** | **1.30** | **0.14** | **1.32** | **0.84** | **0.14** | 0.69 |
| | REF-TX3 | | | 84.7 | **0.22** | *1.32* | **0.14** | **1.33** | **0.84** | **0.14** | 0.69 |
| PM$_{2.5}$ µg.m$^{-3}$ | REF | 8 | 7.2 | 8.4 | **0.17** | *1.33* | **0.51** | *1.68* | **0.72** | **0.23** | 0.41 |
| | REF-TX2 | | | 10.4 | *0.37* | *1.60* | **0.81** | *2.01* | **0.65** | **0.26** | 0.47 |
| | REF-TX3 | | | 12.4 | *0.53* | *1.88* | **1.29** | 2.60 | **0.57** | *0.31* | 0.49 |
| OM µg.m$^{-3}$ | REF | 3 | 4.3 | 1.4 | −0.99 | *0.29* | **2.24** | 7.59 | 0.16 | 0.50 | 0.58 |
| | REF-TX2 | | | 2.6 | −0.46 | *0.54* | **0.78** | *2.51* | *0.37* | *0.35* | 0.59 |
| | REF-TX3 | | | 3.4 | −0.09 | **0.81** | **0.58** | *1.80* | *0.49* | **0.26** | 0.59 |
| C$_5$H$_8$ ppb vol | REF | 2 | 0.29 | 0.09 | −0.89 | *0.25* | 5.95 | *12.27* | 0.15 | 0.54 | 0.54 |
| | REF-TX2 | | | 0.09 | −0.89 | *0.26* | 5.88 | *10.31* | 0.15 | 0.54 | 0.54 |
| | REF-TX3 | | | 0.09 | −0.89 | *0.27* | 5.80 | *9.23* | 0.15 | 0.54 | 0.55 |
| MTs ppb vol | REF | 2 | 0.09 | 0.04 | −0.91 | *0.43* | 12.53 | *2.8e15* | 0.26 | 0.60 | 0.14 |
| | REF-TX2 | | | 0.10 | −0.39 | **0.89** | 6.11 | *4.0e12* | 0.22 | 0.58 | 0.13 |
| | REF-TX3 | | | 0.16 | −0.03 | *1.36* | 5.49 | *1.6e11* | 0.15 | 0.59 | 0.14 |




**Figure 10.** Observed and simulated hourly concentrations of (a) $NO_2$, (b) $O_3$ (c) $PM_{2.5}$ and (d) OM at the Halles station.



The different hypotheses regarding terpene biogenic emissions have low impact on $NO_2$ and $O_3$ concentrations, and all simulations present very similar concentrations and statistical indicators (Table 7). Figure 10a shows good correlation between the $NO_2$ concentrations measured and observed at the Halles site in all simulations, although a few concentration peaks are underestimated. The most strict performance criteria are respected for all statistical indicators for $NO_2$ and for $O_3$. The $O_3$

geometric mean bias is at the limit of the acceptance criteria, because of the overestimation of the low $O_3$ concentrations at night (see Fig. 10b). This overestimation of low $O_3$ concentrations has previously commonly been observed and might be related to model grid resolution (Jang et al., 1995a, b; Liang and Jacobson, 2000; Arunachalam et al., 2006).

For $PM_{2.5}$, the less strict criteria are respected for the three simulations, but the fractional bias (FB) increases with the increase of biogenic terpene emissions. This increase is observed mostly in rural stations. In other words, $PM_{2.5}$ concentrations are

overestimated at rural stations when the terpene biogenic emissions are increased, but the increase of terpene biogenic emissions does not degrade the scores at urban and suburban stations, and it even improves the correlation. A $PM_{2.5}$ concentration peak reaching 80 µg.m$^{-3}$ is observed on July 19 (not shown in Fig. 10c) and is probably due to forest fires in the south-west of France (Menut et al., 2023). Similar to $PM_{2.5}$, the concentrations of the organic fraction of $PM_1$ (organic matter, OM) are strongly influenced by the terpene biogenic emissions hypothesis. While OM concentrations are strongly underestimated in

the REF simulation (fractional bias of $-0.99$), they respect all the less strict criteria in the REF-TX2 and REF-TX3 simulations (fractional bias equal to $-0.46$ and $-0.09$ respectively). As the stations where OM is measured are suburban and urban stations, this goes hand in hand with the better estimate of $PM_{2.5}$ at urban stations (not shown). As shown in Fig. 10c, the effect of modifying biogenic terpene emissions is quite significant, even at the Halles station, which is located in a very dense urban area. This increase of $PM_{2.5}$ concentrations is due to the increase of OM, as shown in Fig. 10d. OM concentrations are

especially high between 18 and 19 June, days with very high temperatures and high biogenic emissions. During this period, the differences between the OM concentrations in the REF, REF-TX2 and REF-TX3 simulations are the largest. The highest the terpene emissions, the better the simulated OM concentration compared to observation, suggesting that it is essential to well represent the terpene emission of suburban areas to well represent the OM concentrations.

Regarding BVOC concentrations, no differences in the three simulations are observed for isoprene ($C_5H_8$) concentrations, as

expected, and the mean concentration tends to be underestimated. Monoterpene concentrations are highly influenced by the biogenic terpene emissions. The higher the biogenic terpene emissions are, the smaller are the fractional biases observed in the simulations ($-0.91$ for REF, $-0.39$ for REF-TX2 and $-0.03$ for REF-TX3) (Table 7). Figures 11a and 11c show the hourly isoprene concentrations simulated and observed at the PRG station (dense urban area) and at the SIRTA station (suburban area), respectively. Isoprene is better represented at SIRTA than at the PRG station, because of the absence of biogenic emissions

inside Paris in REF, REF-TX2 and REF-TX3 simulations.Figures 11b and 11d illustrate the hourly concentrations of monoterpenes simulated and observed at the PRG and SIRTA stations, respectively. Similarly as observed for isoprene, monoterpene concentrations are also strongly underestimated in urban areas (PRG) and better represented at the SIRTA suburban site. This can be justified by the absence of monoterpene biogenic emissions inside Paris, as analyzed in Section 4.2. The observed values in the urban PRG site point out a "regional background" of the monoterpene concentrations around 0.1 ppb.





(a) Isoprene concentrations at the PRG station

(b) Monoterpene concentrations at the PRG station

(c) Isoprene concentrations at the SIRTA station

(d) Monoterpene concentrations at the SIRTA station

**Figure 11.** Observed and simulated hourly concentrations of (a) isoprene and (b) monoterpenes at the PRG station and (c) isoprene and (d) monoterpenes at the SIRTA station.





## 4.2 Impact of biogenic emissions from urban trees on isoprene, monoterpene, ozone, organic matter and PM$_{2.5}$ concentrations

### 4.2.1 Impact of urban-tree biogenic emissions on isoprene and monoterpene concentrations

Comparisons of the hourly concentrations of isoprene and monoterpenes observed and simulated in the reference case (REF-TX2) and with the urban-tree biogenic emissions (bioparis-TX2) at PRG are presented in Fig. 12.

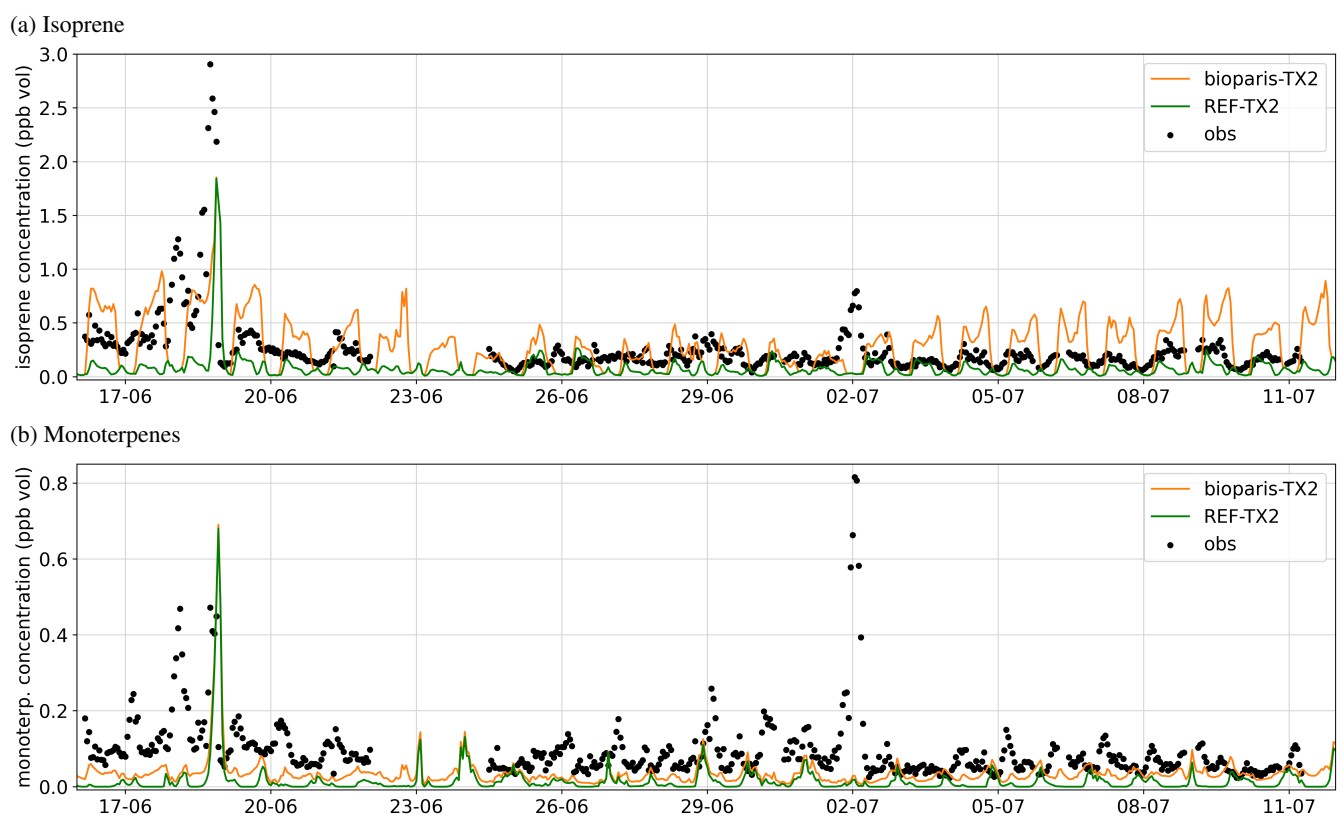

**Figure 12.** Observed and simulated hourly concentrations of (a) isoprene and (b) monoterpenes at the PRG station with (bioparis-TX2) and without (REF-TX2) the bottom-up biogenic emission inventory.

Figure 12a shows that isoprene concentrations simulated at PRG are underestimated in the reference simulation compared to the measurements. The inclusion of the urban-tree biogenic emissions allows a better representation of the isoprene concentrations (decrease of the NAD from 0.57 (REF-TX2) to 0.38 (bioparis-TX2) and increase of the correlation from 0.38 (REF-TX2) to 0.42 (bioparis-TX2)). However, in the bioparis-TX2 simulation, the daytime concentrations are overestimated on June 16, 17, 19, 20 and between July 2 and 10 by about a factor of 1.5, but the concentration peak around the 18$^{th}$ of June is underestimated. At night, non-zero isoprene concentrations are measured, the simulated concentrations are almost zero




because isoprene is emitted only during the day by biogenic emissions and has a short lifetime ($\tau_{OH} \approx 1.5$ h with [OH]$= 10^6$ molecules.cm$^{-3}$ (Atkinson and Arey, 2003b)). Isoprene is also emitted by road-traffic according the VOC speciation used (Theloke and Friedrich, 2007; Baudic et al., 2016), but in the model, traffic emissions are too low at night to represent the measured concentrations. This model to measurement discrepancy could be due to a measurement artefact, or to missing anthropogenic

sources of isoprene at night. In view of the uncertainties in the measurements, the model provides a satisfactory representation of the order of magnitude of isoprene concentrations.

Table 8 presents the averaged isoprene and monoterpene concentrations and the relative impact of bioparis during the 2-month period and heatwaves. Note that the relative difference of concentrations is calculated on an hourly time step and then averaged over the 2-month or heatwave periods. As seen previously, biogenic emissions are driven by environmental variables, in partic-

ular temperature and solar radiation. To determine whether the effect of local biogenic emissions is greater during heatwaves, isoprene concentrations are also compared during these periods. It is especially relevant to quantify this effect because the frequency of these episodes is expected to increase in the future due to climate change (IPCC, 2021). The heatwave periods refers to the averaged concentrations on the following days: June, 15, 16, 17, 18 and July 11, 12, 13, 14, 17, 18, 19. During these periods, high air temperatures and clear sky conditions were observed as shown in Fig. 9. Table 8 shows that at the scale

of the city of Paris, local isoprene emissions significantly increase isoprene concentrations (+1100% on average). The effect of bioparis during the heatwave periods is higher (+2400% on average), because emissions during that period are higher. As the TX2 and TX3 scenarios do not modify isoprene emissions, there is no impact on isoprene concentrations.

The comparison of monoterpene concentrations presented in Fig. 12b shows that monoterpenes are underestimated at PRG in the reference simulation and the addition of the urban-tree emissions strongly increases the monoterpene concentrations. How-

ever, the simulated concentrations still underestimate the observations, even with the bioparis-TX2 scenario, probably because of missing anthropogenic sources (Jo et al., 2023). Like isoprene, Table 8 shows that the addition of monoterpene emissions greatly increases the monoterpene concentrations by $6.4 \times 10^{12}\%$ on average over the 2-month period and by $1.4 \times 10^8\%$ during the heatwave periods. Monoterpene concentrations logically increase when their emissions are multiplied by 2 (TX2) or 3 (TX3), but the simulated concentrations underestimate the measurements. This discrepancy raises the question of potentially

missing local sources of vegetation that emits monoterpenes in the area of measurement.




**Table 8.** Comparison of minimum, mean and maximum isoprene and monoterpene concentrations averaged in Paris for each simulation and relative difference between the bioparis and the reference simulations during the 2 months and the heatwave periods.

| species | simulation | 2-month period | | | Heatwave periods | | |
|---|---|---|---|---|---|---|---|
| | concentration (ppb vol) | min | mean | max | min | mean | max |
| | REF | 0.03 | 0.05 | 0.13 | 0.08 | 0.11 | 0.23 |
| | REF_TX2 | 0.03 | 0.05 | 0.13 | 0.08 | 0.11 | 0.23 |
| | REF_TX3 | 0.03 | 0.05 | 0.13 | 0.09 | 0.12 | 0.22 |
| | bioparis | 0.04 | 0.28 | 0.70 | 0.12 | 0.61 | 1.51 |
| | bioparis_TX2 | 0.04 | 0.28 | 0.68 | 0.12 | 0.61 | 1.48 |
| Isoprene | bioparis_TX3 | 0.04 | 0.27 | 0.67 | 0.12 | 0.60 | 1.45 |
| | Relative difference (%) between | min | mean | max | min | mean | max |
| | bioparis and REF | 40 | 1.1e3 | 7.1e3 | 58 | 2.4e3 | 1.5e4 |
| | bioparis_TX2 and REF_TX2 | 38 | 1.1e3 | 7.1e3 | 53 | 2.3e3 | 1.5e4 |
| | bioparis_TX3 and REF_TX3 | 36 | 1.1e3 | 7.1e3 | 48 | 2.3e3 | 1.5e4 |
| | concentration (ppb vol) | min | mean | max | min | mean | max |
| | REF | 0.005 | 0.009 | 0.12 | 0.009 | 0.016 | 0.19 |
| | REF_TX2 | 0.01 | 0.02 | 0.23 | 0.02 | 0.03 | 0.38 |
| | REF_TX3 | 0.02 | 0.03 | 0.35 | 0.04 | 0.07 | 0.57 |
| | bioparis | 0.007 | 0.02 | 0.12 | 0.01 | 0.03 | 0.21 |
| | bioparis_TX2 | 0.01 | 0.04 | 0.24 | 0.03 | 0.08 | 0.41 |
| Monoterpenes | bioparis_TX3 | 0.03 | 0.07 | 0.36 | 0.05 | 0.13 | 0.61 |
| | Relative difference (%) between | min | mean | max | min | mean | max |
| | bioparis and REF | 3.6 | 6.4e12 | 9.0e13 | 6.3 | 1.4e8 | 1.0e9 |
| | bioparis_TX2 and REF_TX2 | 3.5 | 1.1e13 | 1.5e14 | 6.2 | 1.5e8 | 1.2e9 |
| | bioparis_TX3 and REF_TX3 | 3.4 | 1.7e13 | 2.0e14 | 6.1 | 1.6e8 | 1.6e9 |





### 4.2.2 Impact of urban-tree biogenic emissions on organic matter and particles concentrations

Figure 13, which compares the observed and simulated OM concentrations at PRG, shows that the impact of the urban biogenic emissions is smaller on OM concentrations than on isoprene and monoterpene concentrations. The urban biogenic emissions lead to an increase in OM concentrations on average over Paris during the 2-month period of 4.6% as shown in Table 9. The

increase of OM concentrations is slightly larger when terpene emissions are doubled (+5.6%) and tripled (+6.1%). Due to larger biogenic emissions, the increase in OM concentrations is also larger during the heatwave (+5.4%).

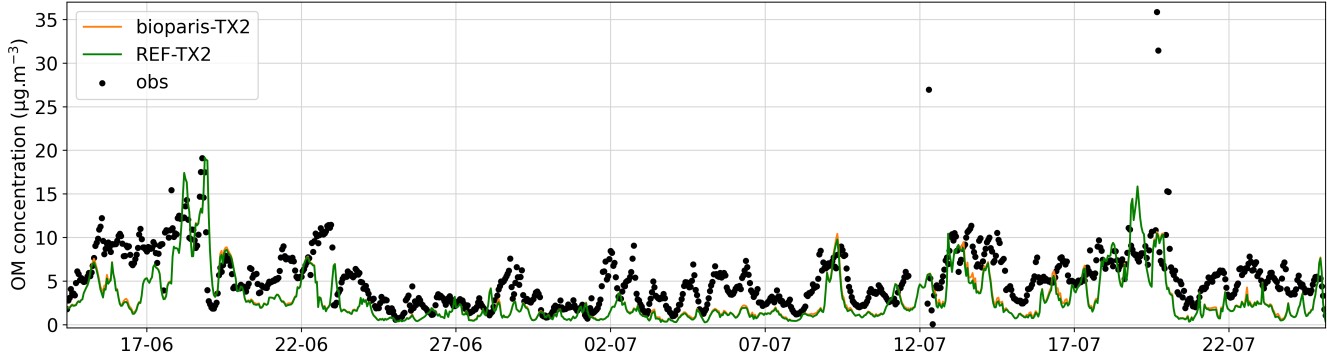

**Figure 13.** Observed and simulated hourly concentrations of organic matter (OM) at the PRG station with (bioparis-TX2) and without (REF-TX2) the urban biogenic emissions inventory.

The impact of the urban biogenic emissions is also less visible on hourly concentrations of $PM_{2.5}$, so the relative differences in OM and $PM_{2.5}$ concentrations are mapped in Figures 14 and 15. The two top panels present the REF-TX2 concentrations and the relative difference between bioparis-TX2 and REF-TX2 concentrations averaged on the 2-month period. The same

maps are presented in the two lower panels but with concentrations averaged on the heatwave periods.





**Figure 14.** Average OM concentrations (µg.m$^{-3}$) simulated by CHIMERE (REF-TX2) during (a) the whole period and (c) the heatwave and relative difference of OM concentrations with the urban-tree biogenic emissions (bioparis-TX2) during (b) the whole period and (d) the heatwave.

Figure 14 shows the spatial variability of the local biogenic emission effect and that the increase in OM concentrations due to emissions from urban trees remains localized over Paris. It is greater in cells with a large tree biomass (Fig. 6), where biogenic emissions are also larger, in particular monoterpenes (Fig. C1) and sesquiterpenes (Fig. C3). This correlation shows that biogenic emissions from urban trees contribute strongly to local OM formation. The increase in OM concentrations is
slightly larger during the heatwaves periods, as shown in Table 9. The impact of urban emissions extends a little further during these periods.





**Figure 15.** Average PM$_{2.5}$ concentrations (µg.m$^{-3}$) simulated by CHIMERE (REF-TX2) during (a) the whole period and (c) the heatwave and relative difference of PM$_{2.5}$ concentrations with the urban-tree biogenic emissions (bioparis-TX2) during (b) the whole period and (d) the heatwave.

The increase in PM$_{2.5}$ concentrations is lower than for OM, but the spatial distribution is similar. The impact remains localized over Paris (+0.6% on average), and is strongest during heatwaves (+1.3%), as shown by the maps in Fig. 15 and Table 9. The increase in PM$_{2.5}$ is larger when monoterpene and sesquiterpene emissions are doubled (TX2) and tripled (TX3) (Table 9). This underlines the importance of terpenes in the formation of particulate matter.





**Table 9.** Comparison of minimum, mean and maximum organic matter (OM) and PM$_{2.5}$ concentrations averaged in Paris for each simulation and relative difference between the bioparis and the reference simulations during the 2 months and the heatwave periods.

| species | simulation | 2-month period | | | Heatwave periods | | |
|---|---|---|---|---|---|---|---|
| | concentration (µg.m$^{-3}$) | min | mean | max | min | mean | max |
| | REF | 1.27 | 1.35 | 1.57 | 2.91 | 3.01 | 3.34 |
| | REF_TX2 | 2.34 | 2.45 | 2.79 | 5.45 | 5.62 | 6.31 |
| | REF_TX3 | 3.45 | 3.58 | 4.14 | 7.94 | 8.16 | 9.27 |
| | bioparis | 1.32 | 1.40 | 1.63 | 2.95 | 3.13 | 3.48 |
| OM | bioparis_TX2 | 2.43 | 2.55 | 2.87 | 5.58 | 5.87 | 6.59 |
| | bioparis_TX3 | 3.57 | 3.74 | 4.26 | 8.15 | 8.55 | 9.69 |
| | Relative difference (%) between | min | mean | max | min | mean | max |
| | bioparis and REF | 0.6 | 4.60 | 11.51 | 0.2 | 5.44 | 14.37 |
| | bioparis_TX2 and REF_TX2 | 0.6 | 5.58 | 15.86 | 0.3 | 6.08 | 18.00 |
| | bioparis_TX3 and REF_TX3 | 0.6 | 6.12 | 18.42 | 0.3 | 6.45 | 20.58 |
| | concentration (µg.m$^{-3}$) | min | mean | max | min | mean | max |
| | REF | 8.48 | 9.25 | 10.85 | 10.83 | 11.64 | 13.36 |
| | REF_TX2 | 9.85 | 10.65 | 12.26 | 14.13 | 14.97 | 16.76 |
| | REF_TX3 | 11.32 | 12.15 | 13.84 | 17.65 | 18.52 | 20.45 |
| | bioparis | 8.54 | 9.31 | 10.91 | 10.85 | 11.78 | 13.53 |
| PM$_{2.5}$ | bioparis_TX2 | 9.98 | 10.77 | 12.38 | 14.18 | 15.25 | 17.16 |
| | bioparis_TX3 | 11.53 | 12.34 | 14.06 | 17.72 | 18.96 | 21.07 |
| | Relative difference (%) between | min | mean | max | min | mean | max |
| | REF and bioparis | 0.12 | 0.64 | 1.60 | 0.09 | 1.25 | 3.09 |
| | REF_TX2 and bioparis_TX2 | 0.20 | 1.12 | 3.06 | 0.14 | 2.06 | 5.78 |
| | REF_TX3 and bioparis_TX3 | 0.25 | 1.55 | 4.52 | 0.17 | 2.69 | 8.29 |





### 4.2.3 Impact on ozone concentrations

Ozone concentrations also increase with the urban-tree biogenic emissions (+1% on average), especially during the heatwave periods (+2.4%). The increase in $O_3$ concentrations is also mostly localized in the Paris city and extends to the Paris suburbs during heatwaves (Fig. 16). Table 10 shows that doubled or tripled monoterpene and sesquiterpene emissions increase ozone

concentrations, but the increase is relatively lower than for OM and $PM_{2.5}$. This suggests that ozone formation is less sensitive to monoterpene and sesquiterpene emissions, which mostly impact the formation of organic matter.

**Figure 16.** Average $O_3$ concentrations (ppb vol) simulated by CHIMERE (REF-TX2) during (a) the whole period and (c) the heatwave and relative difference of $O_3$ concentrations with the urban-tree biogenic emissions (bioparis-TX2) during (b) the whole period and (d) the heatwave.





**Table 10.** Comparison of minimum, mean and maximum ozone concentrations averaged in Paris for each simulation and relative difference between the bioparis and the reference simulations during the 2 months and the heatwave periods.

| simulation | 2-month period | | | Heatwave periods | | |
|---|---|---|---|---|---|---|
| concentration (ppb vol) | min | mean | max | min | mean | max |
| REF | 34.92 | 41.54 | 43.81 | 40.96 | 46.94 | 49.53 |
| REF_TX2 | 35.42 | 42.03 | 44.30 | 41.65 | 47.68 | 50.28 |
| REF_TX3 | 35.87 | 42.46 | 44.73 | 42.27 | 48.34 | 50.95 |
| bioparis | 35.48 | 42.05 | 44.47 | 41.63 | 48.26 | 51.15 |
| bioparis_TX2 | 36.06 | 42.62 | 45.05 | 42.36 | 49.14 | 52.10 |
| bioparis_TX3 | 36.59 | 43.12 | 45.57 | 43.01 | 49.94 | 52.95 |
| Relative difference (%) between | min | mean | max | min | mean | max |
| bioparis and REF | 0.28 | 1.03 | 2.38 | 0.40 | 2.42 | 5.72 |
| bioparis_TX2 and REF_TX2 | 0.32 | 1.17 | 2.67 | 0.48 | 2.65 | 6.21 |
| bioparis_TX3 and REF_TX3 | 0.35 | 1.30 | 2.94 | 0.45 | 2.87 | 6.66 |

The urban biogenic emissions mainly increase $O_3$ concentrations during the day, as the concentrations of biogenic species are higher and $O_3$ is formed during daytime influenced by solar radiation. The impact of the biogenic bottom-up inventory on maximal daily ozone concentrations (8 h moving average) is also evaluated, as this value is used in the French air quality standards (LCSQA, 2016). The bottom-up inventory increases, in average during the 2 months, by around 0.6% the ozone maximal 8 h concentrations and by 1.2% during heatwaves in all scenarios. The maximal impact goes from 4.0% to 4.8% according to the biogenic emission factors scenario on average over the 2-month period and from 7.6 to 8.5% during heatwaves.





## 5 Conclusions

To conclude, trees naturally emit BVOCs, which can lead to the formation of secondary pollutants such as ozone and secondary
organic aerosols. The impact of urban trees on pollutant concentrations is not taken into account in regional air-quality models.
To estimate this impact, an inventory of biogenic emissions from urban trees has been developed using a bottom-up approach.
First, the location and characteristics of the urban trees were obtained from the tree database of the Paris city (Municipality
of Paris, 2023). This information was combined with allometric equations developed for urban trees in the United-States (US)
(McPherson et al., 2016) to compute the leaf dry biomass used in the emissions model. Tree-species emission factors from
MEGAN model were used to compute emissions of various BVOC by each tree species per leaf biomass amount and at stan-
dard conditions. Then, emissions were modulated by the temperature and radiation with activity factors from Guenther et al.
(1995, 2012) and the meteorological variables simulated by WRF. Biogenic emissions were then integrated in the CHIMERE
grid and complement the regional biogenic emissions computed with the land-use approach. The order of magnitude of emis-
sions are consistent between the urban and the regional biogenic emissions.

Air-quality simulations over the Paris region during the 2 months of June and July 2022 lead to simulated $NO_2$, $O_3$, and
$PM_{2.5}$ concentrations that are globally consistent with measurements. OM, isoprene and monoterpene concentrations are un-
derestimated but they increase when emissions from urban trees are taken into account. Over Paris city, urban trees induce a
significant increase in OM concentrations of 4.6% on average over the two months and of 5.4% during the heatwave periods.
This increase can reach 11.5% locally on average over the two months and 14.4% during the heatwave period. The increase in
OM concentrations is sensitive to monoterpene and sesquiterpene emissions and remains localized over Paris city where the
urban trees are located. $O_3$ concentrations also slightly increase due to the urban-tree emissions by 1.0% on average over the
2 months and by 2.4% during the heatwaves. This increase can locally reach 2.4% on average over the two months and 5.7%
during the heatwaves. The increase in $O_3$ concentrations during the heatwave periods extends to the Paris suburbs, further than
for OM. These values correspond to temporal averages but the effect of urban emissions on OM, $PM_{2.5}$ and $O_3$ are higher
during the day time when biogenic emissions and photolysis occur, aggravating $O_3$ peaks during heatwaves. This shows that
urban tree emissions have a large impact on air quality, and low emitting tree species should be favored in cities.

OM concentrations are particularly sensitive to terpene emissions. It is essential to better estimate terpene emission factors
of urban and suburban trees. Furthermore, it should be noted that part of the urban vegetation (in private areas) and of the
suburban vegetation were not taken into account in this study, as the tree inventory is only available for the public trees of Paris
city. The effect of urban and suburban trees on air quality is therefore probably underestimated. Tree inventories should be set
up systematically in more cities and their suburbs. This could be completed with methods for characterizing urban vegetation
using aerial images, for example. This methodology for building a BVOCs bottom-up inventory could be easily applied to
other cities that have a tree inventory.

Further work would involve improving the estimation of the tree-scale biogenic emissions by improving the spatial resolution
of the meteorological fields. Speciation of monoterpenes and oxygenated VOCs emitted into model chemical species is as-
sumed to be identical for each tree species. A speciation of monoterpenes according to the tree species, as done in Steinbrecher



et al. (2009), could be introduced. However as this speciation does not include all the tree species found in Paris, the speciation should be enriched with other data.

*Code availability.* The version of WRF-CHIMERE code used here is available on request.

*Data availability.* ACSM data measured at the PRG site are available in the Aeris datacenter: Di Biagio et al. (2023), ACROSS_LISA_PRG_ACSM-nrPM1comp_L2, in preparation. [Dataset]. Aeris.

PTR-MS data measured at the PRG site are available in the Aeris datacenter: https://www.aeris-data.fr/.

PTR-MS data measured at the SIRTA station are available in the ACTRIS database: https://ebas-data.nilu.no and in the IPSL data catalog: Simon, L., Gros, V., Truong, F., Sarda-Esteve, R., and Kalalian, C.: PTR-MS measurements in 2020–2021, IPSL Data Catalog [dataset], 495   https://doi.org/10.14768/f8c46735-e6c3-45e2-8f6f-26c6d67c4723, 2022a.

Hourly $NO_2$, $O_3$ and $PM_{2.5}$ concentrations measured at the Paris Chatelet/Halles station are available on the Airparif's Open Data Portal: https://data-airparif-asso.opendata.arcgis.com/. Regional emissions inventory and organic matter data for the Halles site are available on request.



**Appendix A: Detailed description of the experimental measurements**

**A1    Measurements performed at SIRTA**

Isoprene was measured with a gas-chromatograph equipped with a Flame Ionisation Detector (GC-FID), AIRMOVOC C2-C6 (Chromatotec, Saint Antoine, France). The instrument is described in detailed Gros et al. (2011). Calibration was performed with a NPL (National Physics Laboratory, Teddigton, UK) standard. Uncertainty are estimated to be less than 15%. Monoterpenes were measured at SIRTA using a Proton-Transfer-Reaction Quadrupole Mass Spectrometer (PTR-Q-MS) from Ionicon (Innsbrueck, Austrria) with a time resolution of 5 min. This instrument was implemented at SIRTA for long-term measurements early 2020 and its operating conditions are described in Simon et al. (2023). The ambient air was sampled at 15m, 1-hour blank measurements were performed every 13 hours, and calibrations every month with a NPL standard containing $\alpha$-pinene. Monoterpenes were measured at the mass-to-charge ratio (m/z) 137, and the associated uncertainties for the period of June-July 2022 were 32%, while the mean detection limit was of 25 ppt.

**A2    Measurements performed at PRG**

Gas and aerosol sampling at PRG site are performed at 30 m above ground layer. VOCs were measured at PRG site using a PTR-ToF-MS (PTR 4000x2, Ionicon Analytik, Austria) equipped with a CHARON inlet, already extensively described elsewhere (Jordan et al., 2009; Eichler et al., 2015; Müller et al., 2017; Leglise et al., 2019). The instrument has been programmed to automatically switch between gas and particle phases, and was working at 2.6 mbar and at E/N=120 Td. Gas was sampled at the top of a 7th floor building through a 12 m long Teflon tubing, with a 17.5 mm inner diameter. The flow in this main line was fixed at 40 L min-1 until a glass manifold where all gas phase instruments sampled ambient air. Sensitivity and background have been regularly controlled during the course of the experiment using pure nitrogen cylinder (99.99999% purity, Linde) and a certified gas standard (containing 10 VOCs at 100 ppb, NPL) providing quantitative measurement with an uncertainty typically in the order of 10 ppt.

**A3    Measurements performed at the Halles**

In the Halles station, $NO_2$ concentrations are measured by chemiluminescence detection with a AC32M, analyzer from EN-VEA (formerly Environnement SA), with a measurement uncertainty of 10%. $O_3$ concentrations are measured by Ultraviolet (UV) photometry with a O3 42e analyzer from ENVEA, with a measurement uncertainty of 11%. $PM_{2.5}$ are measured with a FIDAS 200 analyzer from PALAS, certified technically compliant by the Laboratoire Central de la Surveillance de la Qualité de l' Air (LCSQA) for continuous, real-time regulatory monitoring of $PM_{10}$ and $PM_{2.5}$ fractions based on the optical detection of light scattered by aerosols (Lorenz-Mie solution). The uncertainties associated with measurement are estimated to 9%. More information on the certified devices for regulatory air quality measurement are available here (in French): https://www.lcsqa.org/system/files/media/documents/Liste%20appareils%20conforme%20mesure%20_qualit%C3%A9%20air%20M%C3%A0J_13-05-20_v2_0.pdf



**Table A1.** List of Airparif stations with the species measured and used in this study.

| station | location | type | species measured |
|---|---|---|---|
| PARIS 1er Les Halles | 48.862128° N, 2.3446227° E | urban background | $NO_2$, $O_3$, $PM_{2.5}$, OM |
| PARIS 7eme | 48.8571944° N, 2.2932778° E | urban background | $NO_2$ |
| PARIS 12eme | 48.8371944° N, 2.3938056° E | urban background | $NO_2$ |
| PARIS 13eme | 48.8284722° N, 2.3595583° E | urban background | $NO_2$, $O_3$ |
| PARIS 15eme | 48.8303889° N, 2.2698861° E | urban background | $NO_2$ |
| PARIS 18eme | 48.8917278° N, 2.345575° E | urban background | $NO_2$, $O_3$, $PM_{2.5}$ |
| AUBERVILLIERS | 48.9039444° N, 2.3847222° E | urban background | $NO_2$ |
| ARGENTEUIL | 48.8278324° N, 2.3805391° E | urban background | $NO_2$ |
| BOBIGNY | 48.9024111° N, 2.4526167° E | urban background | $NO_2$, $PM_{2.5}$ |
| CHAMPIGNY-SUR-MARNE | 48.816692° N, 2.516669° E | urban background | $NO_2$, $O_3$ |
| EVRY | 48.8276389° N, 2.3267111° E | urban background | $NO_2$ |
| LOGNES | 48.8403167° N, 2.6346611° E | urban background | $NO_2$, $O_3$ |
| MONTGERON | 48.7065833° N, 2.4570833° E | urban background | $NO_2$, $O_3$ |
| NEUILLY-SUR-SEINE | 48.8813333° N, 2.2773167° E | urban background | $NO_2$, $O_3$ |
| GENNEVILLIERS | 48.9298219° N, 2.291413° E | urban background | $NO_2$, $PM_{2.5}$ |
| VITRY-SUR-SEINE | 48.7756628° N, 2.374005° E | urban background | $NO_2$, $O_3$, $PM_{2.5}$ |
| GONESSE | 48.9908583° N, 2.4447722° E | suburban background | $NO_2$, $PM_{2.5}$ |
| MANTES-LA-JOLIE | 48.996225° N, 1.7032972° E | suburban background | $NO_2$, $O_3$ |
| MELUN | 48.5281028° N, 2.6539472° E | suburban background | $NO_2$, $O_3$ |
| FONTAINEBLEAU FOREST | 48.4562391° N, 2.6793973° E | rural | $NO_2$, $O_3$, $PM_{2.5}$ |
| SAINT-MARTIN-DU-TERTRE | 49.1082856° N, 2.1531876° E | rural | $O_3$, $PM_{2.5}$ |



**Appendix B: Characteristics of trees in the Paris Tree database**

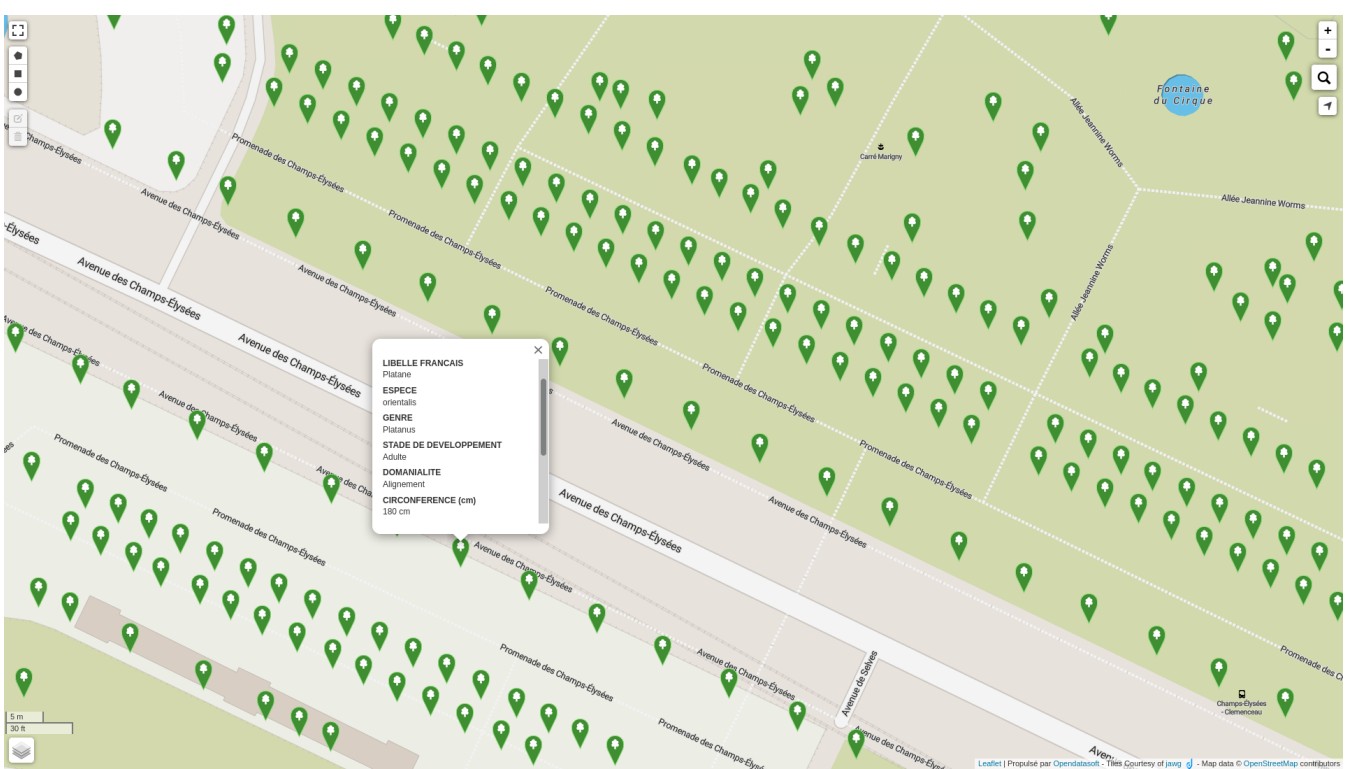

**Figure B1.** Screenshot of the Paris tree database near Avenue des Champs-Élysées (Municipality of Paris, 2023).





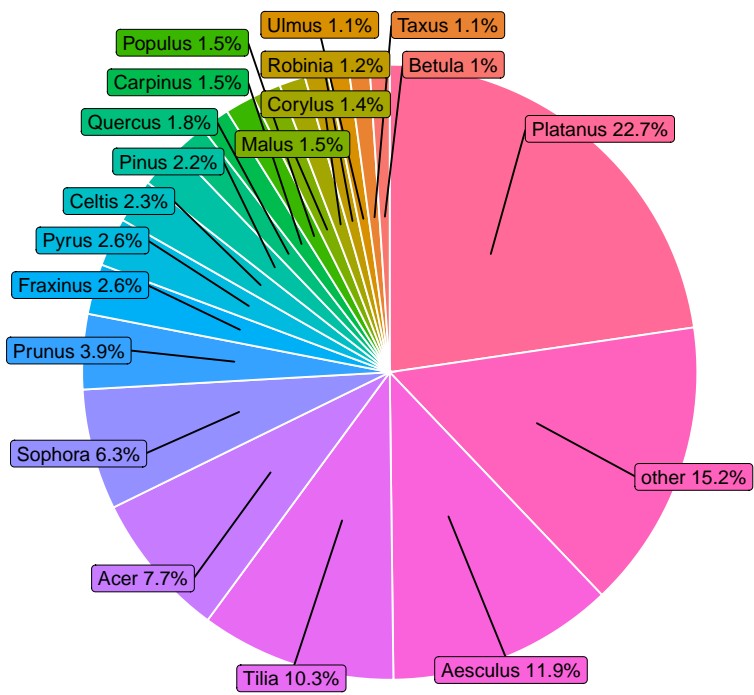

**Figure B2.** Proportion (%) of each tree genus in Paris (only genus with $P > 1\%$ are shown, the rest of the trees are in the "other" category).

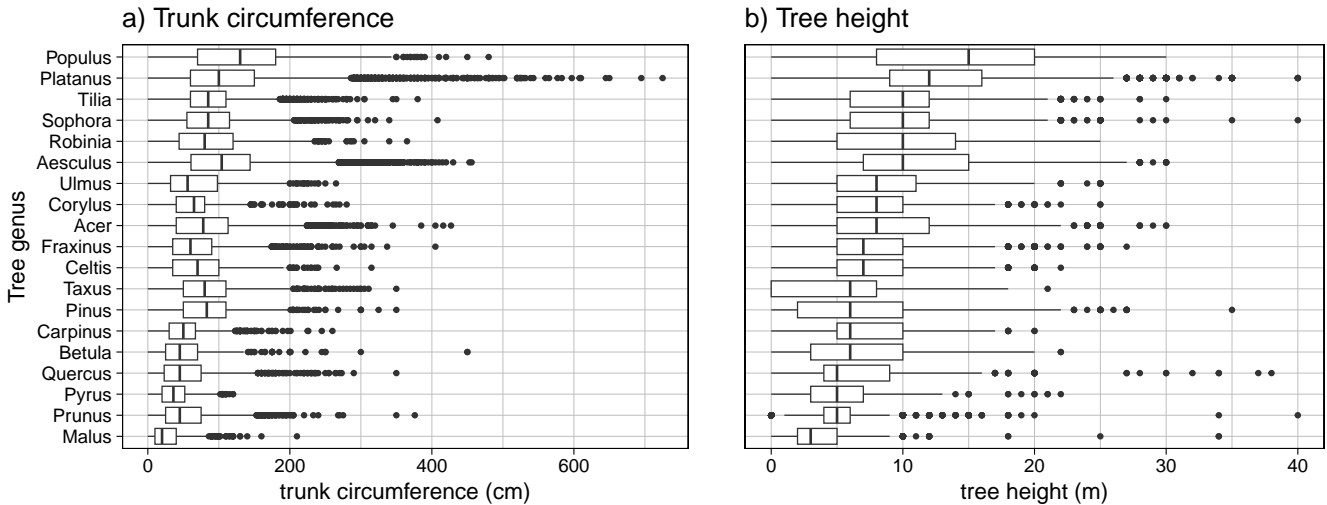

**Figure B3.** Boxplot of the (a) trunk circumference and (b) tree height of the most dominant tree genus.





## Appendix C: BVOC emissions

**Table C1.** Aggregation matrix of emitted MEGAN v3.2 species to MELCHIOR2 species.

| MEGANv3.2/ MELCHIOR2 species | C5H8 | APINEN | BPINEN | LIMONE | TERPEN | OCIMEN | HUMULE | NO | CO | CH3OH | C2H4 | CH3CHO | CH3COE | MEMALD |
|---|---|---|---|---|---|---|---|---|---|---|---|---|---|---|
| ISOP | 1.0 | 0.0 | 0.0 | 0.0 | 0.0 | 0.0 | 0.0 | 0.0 | 0.0 | 0.0 | 0.0 | 0.0 | 0.0 | 0.0 |
| MBO | 0.0 | 0.0 | 0.0 | 0.0 | 0.0 | 0.0 | 0.0 | 0.0 | 0.0 | 0.0 | 0.0 | 0.0 | 0.0 | 1.0 |
| MT_PINE | 0.0 | 1.0 | 0.0 | 0.0 | 0.0 | 0.0 | 0.0 | 0.0 | 0.0 | 0.0 | 0.0 | 0.0 | 0.0 | 0.0 |
| MT_ACYC | 0.0 | 0.0 | 0.0 | 0.0 | 0.0 | 1.0 | 0.0 | 0.0 | 0.0 | 0.0 | 0.0 | 0.0 | 0.0 | 0.0 |
| MT_CAMP | 0.0 | 0.47 | 0.53 | 0.0 | 0.0 | 0.0 | 0.0 | 0.0 | 0.0 | 0.0 | 0.0 | 0.0 | 0.0 | 0.0 |
| MT_SABI | 0.0 | 0.4 | 0.0 | 0.6 | 0.0 | 0.0 | 0.0 | 0.0 | 0.0 | 0.0 | 0.0 | 0.0 | 0.0 | 0.0 |
| MT_AROM | 0.0 | 1.0 | 0.0 | 0.0 | 0.0 | 0.0 | 0.0 | 0.0 | 0.0 | 0.0 | 0.0 | 0.0 | 0.0 | 0.0 |
| NO | 0.0 | 0.0 | 0.0 | 0.0 | 0.0 | 0.0 | 0.0 | 1.0 | 0.0 | 0.0 | 0.0 | 0.0 | 0.0 | 0.0 |
| SQT_HR | 0.0 | 0.0 | 0.0 | 0.0 | 0.0 | 0.0 | 1.0 | 0.0 | 0.0 | 0.0 | 0.0 | 0.0 | 0.0 | 0.0 |
| SQT_LR | 0.0 | 0.0 | 0.0 | 0.0 | 0.0 | 0.0 | 1.0 | 0.0 | 0.0 | 0.0 | 0.0 | 0.0 | 0.0 | 0.0 |
| MEOH | 0.0 | 0.0 | 0.0 | 0.0 | 0.0 | 0.0 | 0.0 | 0.0 | 0.0 | 1.0 | 0.0 | 0.0 | 0.0 | 0.0 |
| ACTO | 0.0 | 0.0 | 0.0 | 0.0 | 0.0 | 0.0 | 0.0 | 0.0 | 0.0 | 0.0 | 0.0 | 0.0 | 1.0 | 0.0 |
| ETOH | 0.0 | 0.0 | 0.0 | 0.0 | 0.0 | 0.0 | 0.0 | 0.0 | 0.0 | 0.0 | 0.0 | 1.0 | 0.0 | 0.0 |
| ACID | 0.0 | 0.0 | 0.0 | 0.0 | 0.0 | 0.0 | 0.0 | 0.0 | 0.0 | 0.0 | 0.0 | 1.0 | 0.0 | 0.0 |
| LVOC | 0.0 | 0.0 | 0.0 | 0.0 | 0.0 | 0.0 | 0.0 | 0.0 | 0.0 | 0.0 | 0.36 | 0.0 | 0.64 | 0.0 |
| OXPROD | 0.0 | 0.0 | 0.0 | 0.0 | 0.0 | 0.0 | 0.0 | 0.0 | 0.0 | 0.0 | 0.0 | 0.9 | 0.1 | 0.0 |
| STRESS | 0.0 | 0.0 | 0.0 | 0.0 | 0.0 | 0.0 | 0.0 | 0.0 | 0.0 | 0.0 | 1.0 | 0.0 | 0.0 | 0.0 |
| OTHER | 0.0 | 0.0 | 0.0 | 0.0 | 0.0 | 0.0 | 0.0 | 0.0 | 0.0 | 0.0 | 0.0 | 0.0 | 1.0 | 0.0 |
| CO | 0.0 | 0.0 | 0.0 | 0.0 | 0.0 | 0.0 | 0.0 | 0.0 | 1.0 | 0.0 | 0.0 | 0.0 | 0.0 | 0.0 |



**Figure C1.** Comparison of the 2 months averaged monoterpene emissions computed with (a) the "bottom-up" inventory and (b) with the land-cover approach in CHIMERE over Île-de-France and Greater Paris.





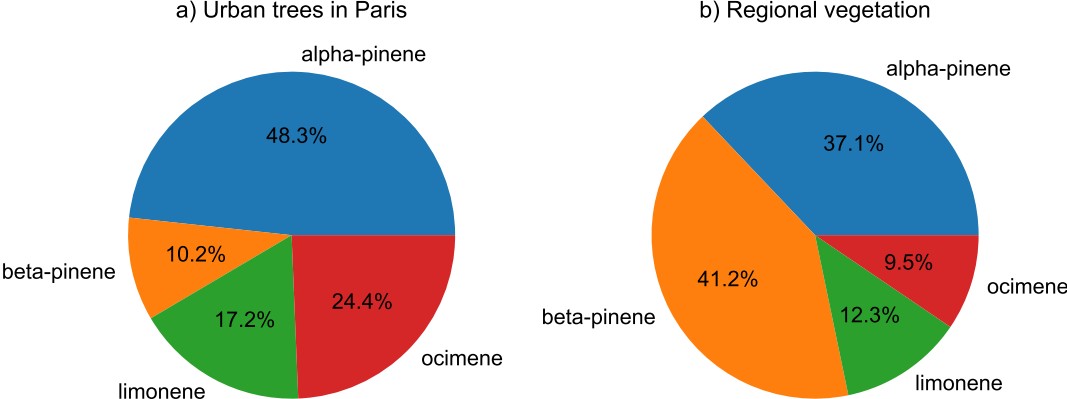

**Figure C2.** Distribution of monoterpene species emitted, summed over the two months (a) for the urban trees in Paris computed with the bottom-up inventory and (b) for the vegetation over Île-de-France region computed with the land-use approach.



**Figure C3.** Comparison of the 2 months sesquiterpene emissions computed with (a) the "bottom-up" inventory and (b) with the land-cover approach in CHIMERE over Île-de-France and Greater Paris.

**Figure C4.** Comparison of the 2 months nitrite oxide (NO) emissions computed with (a) the "bottom-up" inventory and (b) with the land-cover approach in CHIMERE over Île-de-France and Greater Paris.

**Figure C5.** Comparison of the 2 months carbon monoxide (CO) emissions computed with (a) the "bottom-up" inventory and (b) with the land-cover approach in CHIMERE over Île-de-France and Greater Paris.





**Figure C6.** Comparison of the 2 months other VOC (OVOC) emissions computed with (a) the "bottom-up" inventory and (b) with the land-cover approach in CHIMERE over Île-de-France and Greater Paris.





## Appendix D: Definition of the statistical indicators

To compare the simulation results to measured data, classical statistical indicators are computed where $obs_i$ and $sim_i$ are

respectively the observed and simulated hourly concentrations. $n$ is the total number of concentrations and $\overline{obs}$ and $\overline{sim}$ are the

average observed and simulated concentrations.

- Root Mean Square Error (same unit as the concentration):

$$RMSE = \sqrt{\frac{1}{n} \sum_{i=1}^{n} (obs_i - sim_i)^2}. \tag{D1}$$

- Normalized Mean Square Error (dimensionless):

$$NMSE = \frac{\sum_{i=1}^{n} (obs_i - sim_i)^2}{\sum_{i=1}^{n} obs_i \times \sum_{i=1}^{n} sim_i}. \tag{D2}$$

- Normalized Absolute Difference (dimensionless):

$$NAD = \frac{\sum_{i=1}^{n} |obs_i - sim_i|}{\sum_{i=1}^{n} obs_i + \sum_{i=1}^{n} sim_i}. \tag{D3}$$

- Mean Fractional Error (dimensionless):

$$MFE = \frac{1}{n} \sum_{i=1}^{n} \frac{|sim_i - obs_i|}{obs_i}. \tag{D4}$$

- Mean Fractional Bias (same unit as the concentration):

$$MFB = \frac{1}{n} \sum_{i=1}^{n} |sim_i - obs_i|. \tag{D5}$$

- Bias (same unit as the concentration):

$$Bias = \frac{1}{n} \sum_{i=1}^{n} (sim_i - obs_i). \tag{D6}$$

- Fractional Bias (dimensionless):

$$FB = 2 \times \frac{\sum_{i=1}^{n} sim_i - \sum_{i=1}^{n} obs_i}{\sum_{i=1}^{n} obs_i + \sum_{i=1}^{n} sim_i} \tag{D7}$$

- Geometric Mean Bias (dimensionless):

$$MG = \exp\left[ \frac{1}{n} \sum_{i=1}^{n} \ln(sim_i) - \frac{1}{n} \sum_{i=1}^{n} \ln(obs_i) \right] \tag{D8}$$



- Correlation coefficient (dimensionless):

$$R = \frac{\sum\limits_{i=1}^{n} \left[ \left(sim_i - \overline{sim}\right) \left(obs_i - \overline{obs}\right) \right]}{\sqrt{\sum\limits_{i=1}^{n} \left(sim_i - \overline{sim}\right)^2 \sum\limits_{i=1}^{n} \left(obs_i - \overline{obs}\right)^2}}. \tag{D9}$$

- Geometric Variance (dimensionless):

$$VG = \exp\left[ \frac{1}{n} \sum_{i=1}^{n} \left(\ln\left(obs_i\right) - \ln\left(sim_i\right)\right)^2 \right] \tag{D10}$$

- Factor of 2 (dimensionless):

$$FAC2 = \text{total fraction where } 0.5 < \frac{sim_i}{obs_i} < 2.0 \tag{D11}$$

*Author contributions.* KS and AM were responsible for the conceptualization. AM, AT and KS developed the tree biogenic emission inven-
tory. LL, SP, KS, FC and MV prepared the input data for the WRF-CHIMERE model and LL performed the simulations. AM, KS , LL and
SP performed the formal analysis. KS was responsible for the supervision. JV provided the regional and traffic emission inventory. AM and
LL conducted the visualization. The experimental data were provided by AB and JV for the Airparif sites, by AG, CDB, BD, JK, MS, VM
and CC for the PRG site and by VG, JEP, CK and LS for the SIRTA site. AM, LL and KS wrote the original draft, and all authors reviewed
it. KS and AT were responsible for the funding acquisition related to the modeling study.

*Competing interests.* At least one of the (co-)authors is a member of the editorial board of Atmospheric Chemistry and Physics, and the
authors have no other competing interests to declare.

*Acknowledgements.* This work benefited from the French state aid managed by the sTREEt ANR project (ANR-19-CE22-0012), and by
the ANR under the "Investissements d'avenir" program (ANR-11-IDEX-0004-17-EURE-0006) with support from IPSL/Composair. The
measurements at the PRG site have been supported by the ACROSS project. The ACROSS project has received funding from the French
National Research Agency (ANR) under the investment program integrated into France 2030, with the reference ANR-17-MPGA-0002, and
it was supported by the French National program LEFE (Les Enveloppes Fluides et l'Environnement) of the CNRS/INSU (Centre National de
la Recherche Scientifique/Institut National des Sciences de L'Univers). Contributions to measurements at PRG by A. Bauville, M. Cazaunau,
L. Hawkins, D. Pronovost, A. Bergé, L. Di Antonio, F. Maisonneuve, C. Gaimoz, and S. Chevaillier are gratefully acknowledged.



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
