# Peer review of "Significant impact of urban-tree biogenic emissions on air quality estimated by a bottom-up inventory and chemistry-transport modeling"

_EGUsphere, 2023_

## Author Response (AR1)

**Reply on RC1**

Dear reviewer,

Thank you for reviewing this manuscript and for your constructive comments. Please find below a point-by-point answer to your questions. Please note that line, figure, table and equation numbers refer to the original manuscript numbering.
* * *
The manuscript (MS) "Significant impact of urban-tree biogenic emissions on air quality estimated by a bottom-up inventory and chemistry-transport modelling" by Maison et al. is an interesting study about the potential impact of BVOC tree emissions on air quality in Paris. The evaluation is based on both modelling and monitoring data and a rational use of urban tree inventory. The analysis of two months of data, June-July 2022, shows that $O_3$ and OM are influenced by the additional BVOC emissions from urban trees not considered usually in air quality simulations in urban areas.

The work may be an important contribution to this topic if the authors clarify several scientific issues mentioned below and perform a major revision since the MS is very difficult to follow being written more as a report than as a scientific paper.

Scientific issues:

- mention and discuss the meteorological data (hourly or other data) used for offline estimations of BVOC emissions from trees in Material and Methods section;

**Authors' response:** In order to complete the presentation of the weather simulations performed with WRF model, a description of the simulated temperature and solar radiation used to compute BVOC emissions is added in the Material and Methods section (line 128) and in new Appendix C:

"As expected, the WRF model simulates higher temperatures in urban areas than in rural areas (fields or forests), as shown in Fig. C1, which presents the 2-month average air temperature at 2 m simulated by WRF. The simulated downwards shortwave radiation at ground surface (SW) is also used to compute tree biogenic emissions. It is quite homogeneous over the Paris region, with a 2-month average of 500 $W.m^{-2}$ (daytime) and spatial variations within 5% of the mean."

- show a validation of meteorological data over the whole investigated domain, not only in one location, Sirta;

**Authors' response:** In addition to the validation of meteorological data at SIRTA, hourly meteorological data from seven weather stations operated by Météo France and located in IDF1 domain were collected and used for validation. The results are presented in the new Table E2.

- explain in detail how were used the LAI in REF and bioparis simulations; how LAI was used with the computed urban total tree leaf area (LA in m2); if the first is a satellite data, how the calculations avoid the overlap in bioparis simulation;

**Authors' response:** In the REF simulation, the LAI used for BVOC emission computation is taken from the GLOBCOVER land-use database. In urban cell grids, LAI = 0, so the BVOC emissions are nil, as shown in Figures 7b and c. However, the vegetation of Paris's two major parks (Boulogne wood to the west and Vincennes wood to the east) is visible in GLOBCOVER. Over these two parks, the BVOC emissions are computed with the regional approach (REF). Trees in Boulogne and Vincennes woods are not considered as urban trees in bioparis simulations (Fig. 6). So no additional emissions are computed over these two parks (Fig. 7a and b) and there is no overlap in bioparis simulation. In the bioparis simulations, the emissions calculated using the two approaches are added, as they correspond to vegetation located in different places.

To clarify this point, the lines 257 and 258:

"however, these large woods are considered at the regional scale, so their emissions are calculated using the land-use approach as shown in Fig. 7b."

are replaced by:

"however, the emissions of these large woods are already modeled at the regional scale using the land-use approach. Thus, they are not considered in the Paris tree inventory added in the simulation bioparis, in order to avoid overlapping of emissions (Fig. 6)."

- motivate the choice of the single-layer urban canopy model (UCM) (Kusaka et al., 2001) with respect to other schemes like BEP available in WRF for example;

**Authors' response:** The single-layer urban canopy model was chosen in WRF for its simplicity and because it allows to input anthropogenic heat fluxes for commercial, high and low intensity residential areas. These anthropogenic heat fluxes are crucial to correctly model the heat island effect, as well as the friction velocities above buildings.

They will be of particular importance in following works, where these simulations will be coupled to a street-network model.

The lines 120 to 127 have been reorganized with the addition of the following sentences:

"The UCM was chosen in WRF because it allows to input anthropogenic heat fluxes (AH) for different urban categories: AH is assumed to be 45 $W.m^{-2}$ for commercial areas, 10 $W.m^{-2}$ for high intensity residential and 5 $W.m^{-2}$ for low intensity residential areas, based on Pigeon et al. (2007b) and Sailor et al. (2015). AH are crucial to correctly model the heat island effect, as well as the friction velocities above buildings."

- discuss the reasons/compatibility of CORINE land-use coverage and Noah Land-Surface Model;

**Authors' response:** To explain the use of CORINE Land Cover and the compatibility with Noah LSM, we propose to complete the lines 122 to 125:

"The spatial distribution of each land-use category used in WRF simulations is based on CORINE Land Cover (available at https://doi.org/10.2909/71c95a07-e296-44fc-b22b-415f42acfdf0). It was chosen for its very fine resolution (250 m), since for 1 km resolution simulations over a city, a detailed description of the land-use is required to correctly describe the urban fabric. To ensure the compatibility with the Noah Land-Surface Model, we converted the classification from CORINE Land Cover into MODIS (Moderate Resolution Imaging Spectroradiometer International) categories, following Vogel and Afshari (2020). Three urban categories are employed to differentiate street and building dimensions, as well as heat transfer parameters in commercial, high and low intensity residential areas."

- discuss the possible impact of emission differences between the anthropogenic emissions used in the domains FRA9 and IDF3 and the anthropogenic emissions used in IDF1 domain. It seems that they are completely independent, and this may have a significant effect on BVOC impact on $O_3$ and PM. It will be useful to show the differences for some pollutants like $NO_2$ and $PM_{2.5}$ over IDF1 domain.

**Authors' response:** The emission inventory used in FRA9 and IDF3 is a top-down inventory, whereas the emission inventory used in IDF1 is a bottom-up inventory. A bottom-up inventory allows a more precise spatial distribution of the emissions, and it is therefore chosen for the simulation with the finest scale (IDF1). However, it is not available at a large enough scale to be used in the other simulations (IDF3 and FRA9). Both methodologies are based on the same emission factors, for example COPERT/EEA

for traffic emissions. The simulations of IDF3 are used as boundary conditions of the IDF1 simulation, so we expect that the impact of the differences between the inventory is small for the concentrations simulated in IDF1. Furthermore, BVOC emissions are computed with the same methodology in all domains. The formation of $O_3$ and organic matter of particles in link with the BVOC emissions depends strongly on VOC and NOx emissions, which are very similar in terms of total between the two inventories (2% higher for NOx and 6% smaller for VOC, for the Airparif inventory compared to the EMEP emission inventory in IDF1 domain).

- how are made the maps shown in Figures 6 and 7, and the maps shown in Figures 14, 15 and 16: using interpolation (which type) or nearest point?

**Authors' response:** The maps show the value of the variable (dry biomass in Fig. 6, emission in Fig. 7 or concentration in Fig. 14, 15 and 16), calculated as an average value in each 1 km x 1 km grid cell. No post-processing such as interpolation or nearest point is performed. The following sentence is added line 258:

"Note that except for Fig. 2, all the maps presented in this study represent the average value of the variable in the 1 km x 1 km grid cells without any post-processing."

Major revision of MS structure:

- Sections 3.1.1 and 3.1.3 may be Annexes, in particular, the later one that only shows the theoretical behaviour of the chosen functions for describing temperature and light effects.

**Authors' response:** Section 3.1.1 has been lightened by moving Table 3, Figure 3 and equations (3), (4) and (5) to the Appendix D1. Small equations (2) and (6) are incorporated directly into the text. Figure 4, which illustrates that tree characteristics depend not only on DBH but also on tree species, is left in the section. Section 3.1.2 and 3.1.3 are merged ("Emission factors by tree species and computation of activity factors"), Table 4 and lines 201 to 232, which are elements taken from the literature, are moved to the Appendices D2 and D3.

- Also, the Tables 6 and 7 may be moved to Annexes together with most of the discussion regarding the validation of meteorological and air quality simulations. The relevant results for this MS are presented in Tables 8, 9 and 10.

**Authors' response:** Sections 4.1.1 and 4.1.2 on the validation of meteorological and air-quality simulations have been summarized to present only the major aspects of the validation. Figure 9 and Table 6 have been moved to Appendix E along with the detailed description of the validation of meteorological simulations. Table 7, Figures 10 and 11 have been moved to Appendix F along with the detailed description of the validation of air-quality simulations.

- In the caption of Tables 8, 9 and 10, it is not clear the meaning of "minimum, mean and maximum ….specie… concentrations averaged in Paris". For example, is an average of the minimum values for each grid point within the area of the city or is the minimum of the average on the map? It should be clearly stated in the MS.

**Authors' response:** These values are the minimum/maximum of the (temporal) average on the map. The following sentence is added line 404:

"The min/max columns in the tables correspond to the minimum/maximum values of the time-averaged concentration or relative difference."

- Figures 10 and 11 should include especially the timeseries (TS) for bioparis-TX2 simulation since it is the topic of this study. In particular, for urban stations PRG and Halles. The reader would like to see the comparison of the simulations with trees in urban area with respect to that without trees.

**Authors' response:** Figures 10 and 11 are dedicated to the model validation, and they have been moved to the new Appendix F that details the model validation. The comparisons of the reference simulation and bioparis-TX2 was in Figure 13 for OM at PRG, the comparison at Les Halles has been added to this figure (in panel b, Fig. 9 now). The lines 422-423:

"Figure 13, which compares the observed and simulated OM concentrations at PRG, shows that the impact of the urban biogenic emissions is smaller on OM concentrations than on isoprene and monoterpene concentrations."

are replaced by:

"Figure 9 compares the observed and simulated OM concentrations at the a) PRG and b) Halles stations. It shows that the impact of the urban biogenic emissions in PRG site is smaller on OM concentrations than on isoprene and monoterpene concentrations. The increase in OM concentrations with urban tree biogenic emissions in the Halles site is mainly visible during the heatwaves (Fig. 9b)."

- In all TS figures should be evidenced the heat waves periods by shadowed areas since they are

**Authors' response:** All TS figures have been modified to include an orange shaded area during heatwave periods.

- The conclusions may start from line 465, the text above this line is already in introduction.

**Authors' response:** The lines 454 to 464 of the conclusion are removed and replaced by a less detailed reminder of the study objectives and methods:

"This study presents the development of an inventory of biogenic emissions from urban trees using a bottom-up approach and based on city tree inventory, tree allometric relations and empirical emission equations. The emissions are computed for individual urban trees and integrated into CHIMERE-WRF simulations to quantify the impact of urban trees on pollutant concentrations."

**Reply on RC2**

Dear reviewer,

Thank you for reviewing this manuscript and for your constructive comments. Please find below a point-by-point answer to your questions. Please note that line, figure, table and equation numbers refer to the original manuscript numbering.

——

This is an important topic and worthy of publication. It deserved a lengthy review. Sadly my review was lost from the system and I have no time to write another. I appreciated the care and detail given by the authors, but had a number of questions.

1. I was concerned about whether isolated trees in urban areas would be the same as those in forests. In particular it would also be useful to know about the climate that urban trees would be exposed to in terms of temperature (hotter), relative humidity (drier), sunlight (lower?), wind (lower). Isolated trees in cities may also have a different morphology and might additionally be subject to pruning etc.

**Authors' response:** It is true than the urban micro-climate affects tree functioning and in particular biogenic emissions. Some of these effects are taken into account through the representation of urban meteorology (higher temperatures, lower humidity). Other effects linked to more local meteorological variations, particularly for roadside trees (shading effects of buildings, water stress) should be taken into account in following works.

This point is added to the perspectives at the end of the conclusion (line 488):

"Finally, urbanization may induce very local modifications of climate in streets, with potentially higher temperatures, modified solar radiation due to building shading and water stress if trees are planted in limited soil volume. These processes are not taken into account in this study, where the spatial resolution is typical of regional urban studies, i.e. 1 km x 1 km."

2. I was not sure whether a reader would grasp the meaning of the word significant in the title. Is this "substantial" or "significant". I could see reasons for both, but are the changes determined in the MS substantial or significant. I always take "significant" to mean statistical significance, but was confused about how the word was being used in the MS.

**Authors' response:** We prefer the word "significant" than "substantial". It can be understood in term of statistical significant.

3. The MS introduction rightly mentions the controversial issue whether urban leaves decrease pollutants in cities. However, I thought the conclusion could return to this issue as one wonders from a policy perspective about the balance of planting trees and the relevance of their positive and negative effects. Policy makers usually want to plant more trees (native rather than exotic), so should they be less enthusiastic, or choose low emission varieties?

**Authors' response:** We have completed our recommendation for public policies by adding the sentence line 476:

"In particular, we recommend choosing to plant tree species that emit few terpenes."

4. Are the private tree numbers substantial enough to account for discrepancies in the model/ observation comparisons.

**Authors' response:** The underestimation of monoterpene and OM concentrations can be explained to some extent by the 30% missing private trees in Paris, but more importantly by all the trees in the nearby suburbs. The sentence line 480 in the conclusion is completed with: "and this may contribute to the underestimation of monoterpene and OM concentrations observed."

5. Lines 170-180 what is the logic of the limited number of species detailed here. How were they chosen for discussion?

**Authors' response:** These three tree species have been chosen because they have different allometric functions. It illustrates the diversity of allometric functions. The sentence line 168 is completed with:

"which have different allometric equation forms"

---

## Referee Report (RR1)

I thank the authors for considering my comments&suggestions.

I recommend the publication of the MS as it is now.